# WITCHES' BREW: INDUSTRIAL SCALE DATA POISONING VIA GRADIENT MATCHING

**Jonas Geiping**[*]
Dep. of Electr. Eng. and Computer Science
University of Siegen
`jonas.geiping@uni-siegen.de`

**Liam Fowl**[*]
Department of Mathematics
University of Maryland
`lfowl@umd.edu`

**W. Ronny Huang**
Department of Computer Science
University of Maryland
`wronnyhuang@gmail.com`

**Wojciech Czaja**
Department of Mathematics
University of Maryland
`wojtek@math.umd.edu`

**Gavin Taylor**
Computer Science
US Naval Academy
`taylor@usna.edu`

**Michael Moeller**[†]
Dep. of Elect. Eng. and Computer Science
University of Siegen
`michael.moeller@uni-siegen.de`

**Tom Goldstein**[†]
Department of Computer Science
University of Maryland
`tomg@umd.edu`

## ABSTRACT

Data Poisoning attacks modify training data to maliciously control a model trained on such data. In this work, we focus on targeted poisoning attacks which cause a reclassification of an unmodified test image and as such breach model integrity. We consider a particularly malicious poisoning attack that is both "from scratch" and "clean label", meaning we analyze an attack that successfully works against new, randomly initialized models, and is nearly imperceptible to humans, all while perturbing only a small fraction of the training data. Previous poisoning attacks against deep neural networks in this setting have been limited in scope and success, working only in simplified settings or being prohibitively expensive for large datasets. The central mechanism of the new attack is matching the gradient direction of malicious examples. We analyze why this works, supplement with practical considerations. and show its threat to real-world practitioners, finding that it is the first poisoning method to cause targeted misclassification in modern deep networks trained from scratch on a full-sized, poisoned ImageNet dataset. Finally we demonstrate the limitations of existing defensive strategies against such an attack, concluding that data poisoning is a credible threat, even for large-scale deep learning systems.

## 1 INTRODUCTION

Machine learning models have quickly become the backbone of many applications from photo processing on mobile devices and ad placement to security and surveillance (LeCun et al., 2015). These applications often rely on large training datasets that aggregate samples of unknown origins, and the security implications of this are not yet fully understood (Papernot, 2018). Data is often sourced in a way that lets malicious outsiders contribute to the dataset, such as scraping images from the web, farming data from website users, or using large academic datasets scraped from social media (Taigman et al., 2014). *Data Poisoning* is a security threat in which an attacker makes imperceptible changes to data that can then be disseminated through social media, user devices, or public datasets without being caught by human supervision. The goal of a poisoning attack is to modify the final model to achieve a malicious goal. In this work we focus on targeted attacks

---

[*]Authors contributed equally.
[†]Authors contributed equally.

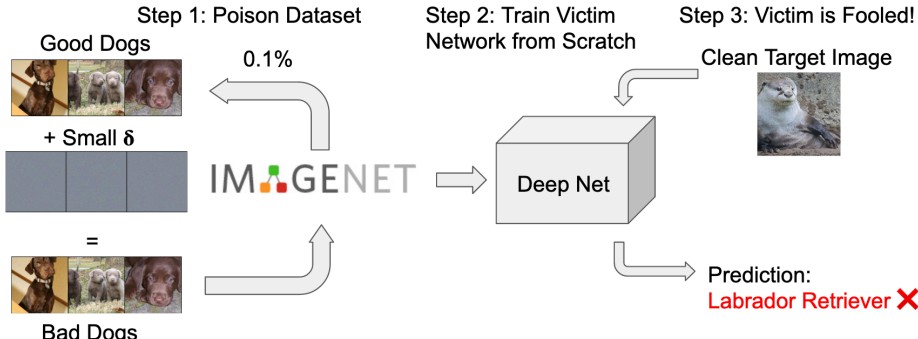

Figure 1: The poisoning pipeline. Poisoned images (labrador retriever class) are inserted into a dataset and cause a newly trained victim model to mis-classify a target (otter) image. We show successful poisons for a threat model where $0.1\%$ of training data is changed within an $\ell_\infty$ bound of $\varepsilon = 8$. Further visualizations of poisoned data can be found in the appendix.

that achieve mis-classification of some predetermined target data as in Suciu et al. (2018); Shafahi et al. (2018), effectively implementing a backdoor that is only triggered for a specific image. Yet, other potential goals of the attacker can include denial-of-service (Steinhardt et al., 2017; Shen et al., 2019), concealment of users (Shan et al., 2020), or introduction of fingerprint information (Lukas et al., 2020). These attacks are applied in scenarios such as social recommendation (Hu et al., 2019), content management (Li et al., 2016; Fang et al., 2018), algorithmic fairness (Solans et al., 2020) and biometric recognition (Lovisotto et al., 2019). Accordingly, industry practitioners ranked data poisoning as the most serious attack on ML systems in a recent survey of corporations (Kumar et al., 2020).

We show that efficient poisoned data causing targeted misclassification can be created even in the setting of deep neural networks trained on large image classification tasks, such as ImageNet (Russakovsky et al., 2015). Previous work on targeted data poisoning has often focused on either linear classification tasks (Biggio et al., 2012; Xiao et al., 2015; Koh et al., 2018) or poisoning of transfer learning and fine tuning (Shafahi et al., 2018; Koh & Liang, 2017) rather than a full end-to-end training pipeline. Attacks on deep neural networks (and especially on ones trained from scratch) have proven difficult in Muñoz-González et al. (2017) and Shafahi et al. (2018). Only recently were targeted attacks against neural networks retrained from scratch shown to be possible in Huang et al. (2020) for CIFAR-10 - however with costs that render scaling to larger datasets, like the ImageNet dataset, prohibitively expensive.

We formulate targeted data poisoning as the problem of solving a *gradient matching* problem and analyze the resulting novel attack algorithm that scales to unprecedented dataset size and effectiveness. Crucially, the new poisoning objective is orders-of-magnitude more efficient than a previous formulation based on on meta-learning (Huang et al., 2020) and succeeds more often. We conduct an experimental evaluation, showing that poisoned datasets created by this method are robustly compromised and significantly outperform other attacks on CIFAR-10 on the benchmark of Schwarzschild et al. (2020). We then demonstrate reliably successful attacks on common ImageNet models in realistic training scenarios. For example, the attack successfully compromises a ResNet-34 by manipulating only $0.1\%$ of the data points with perturbations less than 8 pixel values in $\ell_\infty$-norm. We close by discussing previous defense strategies and how strong differential privacy (Abadi et al., 2016) is the only existing defense that can partially mitigate the effects of the attack.

## 2 RELATED WORK

The task of data poisoning is closely related to the problem of adversarial attacks at test time, also referred to as evasion attacks (Szegedy et al., 2013; Madry et al., 2017), where the attacker alters a target test image to fool an already-trained model. This attack is applicable in scenarios where the attacker has control over the target image, but not over the training data. In this work we are specifically interested in *targeted* data poisoning attacks – attacks which aim to cause a specific target

test image (or set of target test images) to be mis-classified. For example, an attack may cause a certain target image of a otter not part of the training set to be classified as a dog by victim models at test time. This attack is difficult to detect, because it does not noticeably degrade either training or validation accuracy (Shafahi et al., 2018; Huang et al., 2020) and is effectively invisible until it is triggered. From a security standpoint, these attacks break the integrity of a machine learning model and are as such also called *poison integrity* attacks in Barreno et al. (2010) - in contrast to *poison availability* attacks which reduce validation accuracy in general and are not a focus of this work.

In comparison to evasion attacks, targeted data poisoning attacks generally consider a setting where the attacker can modify training data within limits, but cannot modify test data and chooses specific target data a-priori. A related intermediary between data poisoning attacks we consider and evasion attacks are backdoor trigger attacks (Turner et al., 2018; Saha et al., 2019). These attacks involve inserting a trigger – often an image patch – into training data, which is later activated by also applying the trigger to test images. Backdoor attacks require perturbations to both training and test-time data – the more permissive threat model is a trade-off that allows for unknown target images.

Two basic schemes for targeted poisoning are label flipping (Barreno et al., 2010; Paudice et al., 2019), and watermarking (Suciu et al., 2018; Shafahi et al., 2018). In label flipping attacks, an attacker is allowed to change the label of examples, whereas in a watermarking attack, the attacker perturbs the training image, not label, by superimposing a target image onto training images. These attacks can be successful, yet they are easily detected by supervision such as Papernot & McDaniel (2018). This is in contrast to *clean-label* attacks which maintain the semantic labels of data.

Mathematically speaking, data poisoning is a *bilevel* optimization problem (Bard & Falk, 1982; Biggio et al., 2012); the attacker optimizes image pixels to enforce (malicious) criteria on the resulting network parameters, which are themselves the solution to an "inner" optimization problem that minimizes the training objective. Direct solutions to the bilevel problem of data poisoning have been proposed where feasible, for example, SVMs in Biggio et al. (2012) or logistic regression in Demontis et al. (2019). However, direct optimization of the poisoning objective is intractable for deep neural networks because it requires backpropagating through the entire SGD training procedure, see Muñoz-González et al. (2017). As such, the bilevel objective has to be approximated. Recently, MetaPoison (Huang et al., 2020) proposed to approximately solve the bi-level problem based on methods from the meta-learning community (Finn et al., 2017). The bilevel gradient is approximated by backpropagation through several unrolled gradient descent steps. This is the first attack to succeed against deep networks trained from scratch on CIFAR-10 as well as providing transferability to other models. Yet, Huang et al. (2020) uses a complex loss function averaged over a wide range of models trained to different epochs and a single unrolling step necessarily involves both clean and poisoned data, making it roughly as costly as one epoch of standard training. With an ensemble of 24 models, Huang et al. (2020) requires 3 (2 unrolling steps + 1 clean update step) x 2 (backpropagation through unrolled steps) x 60 (first-order optimization steps) x 24 (ensemble of models) equivalent epochs of normal training to attack, as well as ($\sum_{k=0}^{23} k = 253$) epochs of pretraining. All in all, this equates to 8893 training epochs. While this can be mitigated by smart caching and parallelization strategies, unrolled ensembles remain costly.

In contrast to bilevel approaches stand heuristics for data poisoning of neural networks. The most prominent heuristic is *feature collision*, as in Poison Frogs (Shafahi et al., 2018), which seeks to cause a target test image to be misclassified by perturbing training data to collide with the target image in feature space. Modifications surround the target image in feature space with a convex polytope (Zhu et al., 2019) or collection of poisons (Aghakhani et al., 2020) and consider model ensembles (Zhu et al., 2019). These methods are efficient, but designed to attack fine-tuning scenarios where the feature extractor is nearly fixed and not influenced by poisoned data. When applied to deep networks trained from scratch, their performance drops significantly.

## 3 EFFICIENT POISON BREWING

In this section, we will discuss an intriguing weakness of neural network training based on first-order optimization and derive an attack against it. This attack modifies training images that so they produce a *malicious gradient signal* during training, even while appearing inconspicuous. This is done by matching the gradient of the target images within $\ell^\infty$ bounds. Because neural networks are trained by gradient descent, even minor modifications of the gradients can be incorporated into the final model.

This attack compounds the strengths of previous schemes, allowing for data poisoning as efficiently as in Poison Frogs (Shafahi et al., 2018), requiring only a single pretrained model and a time budget on the order of one epoch of training for optimization - but still capable of poisoning the from-scratch setting considered in Huang et al. (2020). This combination allow an attacker to "brew" poisons that successfully attack realistic models on ImageNet.

## 3.1 THREAT MODEL

These discussed components of a clean-label targeted data poisoning attack fit together into the following exemplary threat scenario: Assume a security system that classifies luggage images. An attacker wants this system to classify their particular piece of luggage, the *target* as safe, but can modify only a small part of the training set. The attacker modifies this subset to be *clean-label* poisoned. Although the entire training set is validated by a human observer, the small subset of minorly modified images pass cursory inspection and receive their correct label. The security system is trained on secretly compromised data, evaluated on validation data as normal and deployed. Until the target is evaluated and mis-classified as safe, the system appears to be working fine.

Formally, we define two parties, the *attacker*, which has limited control over the training data, and the *victim*, which trains a model based on this data. We first consider a gray-box setting, where the attacker has knowledge of the model architecture used by their victim. The attacker is permitted to poison a fraction of the training dataset (usually less than 1%) by changing images within an $\ell_\infty$-norm $\varepsilon$-bound (e.g. with $\varepsilon \leq 16$). This constraint enforces *clean-label* attacks, meaning that the semantic label of a poisoned image is still unchanged. The attacker has no knowledge of the training procedure - neither about the initialization of the victim's model, nor about the (randomized) mini-batching and data augmentation that is standard in the training of deep learning models.

We formalize this threat model as bilevel problem for a machine learning model $F(x, \theta)$ with inputs $x \in \mathbb{R}^n$ and parameters $\theta \in \mathbb{R}^p$, and loss function $\mathcal{L}$. We denote the $N$ training samples by $(x_i, y_i)_{i=1}^N$, from which a subset of $P$ samples are poisoned. For notation simplicity we assume the first $P$ training images are poisoned by adding a perturbation $\Delta_i$ to the $i^{th}$ training image. The perturbation is constrained to be smaller than $\varepsilon$ in the $\ell_\infty$-norm. The task is to optimize $\Delta$ so that a set of $T$ target samples $(x_i^t, y_i^t)_{i=1}^T$ is reclassified with the new adversarial labels $y_i^{\mathrm{adv}}$:

$$\min_{\Delta \in \mathcal{C}} \sum_{i=1}^T \mathcal{L}\left(F(x_i^t, \theta(\Delta)), y_i^{\mathrm{adv}}\right) \quad \text{s.t. } \theta(\Delta) \in \arg\min_\theta \frac{1}{N} \sum_{i=1}^N \mathcal{L}(F(x_i + \Delta_i, \theta), y_i). \quad (1)$$

We subsume the constraints in the set $\mathcal{C} = \{\Delta \in \mathbb{R}^{N \times n} : ||\Delta||_\infty \leq \varepsilon, \Delta_i = 0 \ \forall i > P\}$. We call the main objective on the left the *adversarial loss*, and the objective that appears in the constraint on the right is the *training loss*. For the remainder, we consider a single target image ($T = 1$) as in Shafahi et al. (2018), but stress that this is not a general limitation as shown in the appendix.

## 3.2 MOTIVATION

What is the optimal alteration of the training set that causes a victim neural network $F(x, \theta)$ to mis-classify a specific target image $x^t$? We know that the expressivity of deep networks allows them to fit arbitrary training data (Zhang et al., 2016). Thus, if an attacker was unconstrained, a straightforward way to cause targeted mis-classification of an image is to insert the target image, with the incorrect label $y^{\mathrm{adv}}$, into the victim network's training set. Then, when the victim minimizes the training loss they simultaneously minimize the adversarial loss, based on the gradient information about the target image. In our threat model however, the attacker is not able to insert the mis-labeled target. They can, however, still mimic the gradient of the target by creating poisoned data whose training gradient correlates with the adversarial target gradient. If the attacker can enforce

$$\nabla_\theta \mathcal{L}(F(x^t, \theta), y^{\mathrm{adv}}) \approx \frac{1}{P} \sum_{i=1}^P \nabla_\theta \mathcal{L}(F(x_i + \Delta_i, \theta), y_i) \quad (2)$$

to hold for any $\theta$ encountered during training, then the victim's gradient steps that minimize the training loss on the poisoned data (right hand side) will also minimize the attackers adversarial loss on the targeted data (left side).

### 3.3 THE CENTRAL MECHANISM: GRADIENT ALIGNMENT

Gradient magnitudes vary dramatically across different stages of training, and so finding poisoned images that satisfy eq. (2) for all $\theta$ encountered during training is infeasible. Instead we *align* the target and poison gradients in the same direction, that is we minimize their negative cosine similarity. We do this by taking a clean model $F$ with parameters $\theta$, keeping $\theta$ fixed, and then optimizing

$$\mathcal{B}(\Delta, \theta) = 1 - \frac{\left\langle \nabla_\theta \mathcal{L}(F(x^t, \theta), y^{\mathrm{adv}}), \sum_{i=1}^P \nabla_\theta \mathcal{L}(F(x_i + \Delta_i, \theta), y_i) \right\rangle}{\|\nabla_\theta \mathcal{L}(F(x^t, \theta), y^{\mathrm{adv}})\| \cdot \|\sum_{i=1}^P \nabla_\theta \mathcal{L}(F(x_i + \Delta_i, \theta), y_i)\|}. \tag{3}$$

We optimize $\mathcal{B}(\Delta)$ using signed Adam updates with decaying step size, projecting onto $\mathcal{C}$ after every step. This produces an alignment between the averaged poison gradients and the target gradient. In contrast to Poison Frogs, all layers of the network are included (via their parameters) in this objective, not just the last feature layer.

Each optimization step of this attack requires only a *single* differentiation of the parameter gradient w.r.t to its input to compute the objective, instead of requiring the computation and evaluation of several unrolled steps as in MetaPoison to compute the objective. Furthermore, as in Poison Frogs we differentiate through a loss that only involves the (small) subset of poisoned data instead of involving the entire dataset, such that the attack is especially fast if the budget is small. As a side effect this also means that precise knowledge of the full training set is not required, only the poisoned subset and a trained parameter vector $\theta$. Finally, the method is able to create poisons using only a single parameter vector, $\theta$ (like Poison Frogs in fine-tuning setting, but not the case for MetaPoison) and does not require updates of this parameter vector after each poison optimization step.

*Remark.* We find cosine similarity to be exceedingly effective for the classification models considered in this work. However, for small (and especially thin) models there exist a regime in which computing the matching term using squared Euclidean loss between gradients, directly optimizing eq. (2), is a stronger attack. We conduct an ablation study in fig. 12, showing this for thin ResNets.

### 3.4 MAKING ATTACKS THAT TRANSFER AND SUCCEED "IN THE WILD"

A practical and robust attack must be able to poison different random initializations of network parameters and a variety of architectures. To this end, we employ several techniques:

*Differentiable Data Augmentation and Resampling:* Data augmentation is a standard tool in deep learning, and transferable image perturbations must survive this process. At each step minimizing eq. (3), we randomly draw a translation, crop, and possibly a horizontal flip for each poisoned image, then use bilinear interpolation to resample to the original resolution. When updating $\Delta$, we differentiate through this grid sampling operation as in Jaderberg et al. (2015). This creates an attack which is robust to data augmentation and leads to increased transferability.

*Restarts:* The efficiency we gained in section 3.3 allows us to incorporate restarts, a common technique in the creation of evasion attacks (Qin et al., 2019; Mosbach et al., 2019). We minimize eq. (3) several times from random starting perturbations, and select the set of poisons that give us the lowest alignment loss $\mathcal{B}(\Delta)$. This allows us to trade off reliability with computational effort.

---

**Algorithm 1** Poison Brewing via the discussed approach.

---

1: **Require** Pretrained clean network $\{F(\cdot, \theta)\}$, a training set of images and labels $(x_i, y_i)_{i=1}^N$, a target $(x^t, y^{\mathrm{adv}})$, $P < N$ poison budget, perturbation bound $\varepsilon$, restarts $R$, optimization steps $M$
2: **Begin**
3: Select $P$ training images with label $y^{\mathrm{adv}}$
4: **For** $r = 1, \ldots, R$ restarts:
5:      Randomly initialize perturbations $\Delta^r \in \mathcal{C}$
6:      **For** $j = 1, \ldots, M$ optimization steps:
7:          Apply data augmentation to all poisoned samples $(x_i + \Delta_i^r)_{i=1}^P$
8:          Compute the average costs, $\mathcal{B}(\Delta^r, \theta)$ as in eq. (3), over all poisoned samples
9:          Update $\Delta^r$ with a step of signed Adam and project onto $||\Delta^r||_\infty \leq \varepsilon$
10: Choose the optimal $\Delta^*$ as $\Delta^r$ with minimal value in $\mathcal{B}(\Delta^r, \theta)$
11: **Return** Poisoned dataset $(x_i + \Delta_i^*, y_i)_{i=1}^N$

---

*Model Ensembles:* A known approach to improving transferability is to attack an ensemble of model instances trained from different initializations (Liu et al., 2017; Zhu et al., 2019; Huang et al., 2020). However, ensembles are highly expensive, increasing the pre-training cost for only a modest, but stable, increase in performance.

We show the effects of these techniques via CIFAR-10 experiments (see table 1 and section 5.1). To keep the attack within practical reach, we do not consider ensembles for our experiments on ImageNet data, opting for the cheaper techniques of restarts and data augmentation. A summarizing description of the attack can be found in algorithm 1. Lines 8 and 9 of algorithm 1 are done in a stochastic (mini-batch) setting (which we omitted in algorithm 1 for notation simplicity).

## 4 THEORETICAL ANALYSIS

Can gradient alignment cause network parameters to converge to a model with low adversarial loss? To simplify presentation, we denote the adversarial loss and normal training loss of eq. (1) as $\mathcal{L}_{adv}(\theta) =: \mathcal{L}(F((x^t, \theta), y^{adv})$ and $\mathcal{L}(\theta) =: \frac{1}{N} \sum_{i=1}^{N} \mathcal{L}(x_i, y_i, \theta)$, respectively. Also, recall that $1 - \mathcal{B}(\Delta, \theta^k)$, defined in eq. (3), measures the cosine similarity between the gradient of the adversarial loss and the gradient of normal training loss. We adapt a classical result of Zoutendijk (Nocedal & Wright, 2006, Thm. 3.2) to shed light on why data poisoning can work even though the victim only performs standard training on a poisoned dataset:

**Proposition 1** (Adversarial Descent). *Let $\mathcal{L}_{adv}(\theta)$ be bounded below and have a Lipschitz continuous gradient with constant $L > 0$ and assume that the victim model is trained by gradient descent with step sizes $\alpha_k$, i.e. $\theta^{k+1} = \theta^k - \alpha_k \nabla \mathcal{L}(\theta^k)$. If the gradient descent steps $\alpha_k > 0$ satisfy*

$$\alpha_k L < \beta \left(1 - \mathcal{B}(\Delta, \theta^k)\right) \frac{||\nabla \mathcal{L}(\theta^k)||}{||\nabla \mathcal{L}_{adv}(\theta^k)||} \tag{4}$$

*for some fixed $\beta < 1$, then $\mathcal{L}_{adv}(\theta^{k+1}) < \mathcal{L}_{adv}(\theta^k)$. If in addition $\exists \varepsilon > 0$, $k_0$ so that $\forall k \geq k_0$, $\mathcal{B}(\Delta, \theta^k) < 1 - \varepsilon$, then*

$$\lim_{k \to \infty} ||\nabla \mathcal{L}_{adv}(\theta^k)|| \to 0. \tag{5}$$

*Proof.* See supp. material. □

Put simply, our poisoning method aligns the gradients of training loss and adversarial loss. This enforces that the gradient of the main objective is a descent direction for the adversarial objective, which, when combined with conditions on the step sizes, causes a victim to unwittingly converge to a stationary point of the adversarial loss, i.e. optimize *the original bilevel objective* locally.

The strongest assumption in Proposition 1 is that gradients are almost always aligned, $\mathcal{B}(\Delta, \theta^k) < 1 - \epsilon, k \geq k_0$. We directly maximize alignment during creation of the poisoned data, but only for a selected $\theta^*$, and not for all $\theta^k$ encountered during gradient descent from any possible initialization. However, poison perturbations made from one parameter vector, $\theta$, can transfer to other parameter vectors encountered during training. For example, if one allows larger perturbations, and in the limiting case, unbounded perturbations, our objective is minimal if the poison data is identical to the target image, which aligns training and adversarial gradients at every $\theta$ encountered. Empirically, we see that the proposed "poison brewing" attack does indeed increase gradient alignment. In fig. 2, we see that in the first phase of training all alignments are positive, but only the poisoned model maintains a positive similarity for the adversarial target-label gradient throughout training. The clean model consistently shows that these angles are negatively aligned - i.e. normal training on a clean

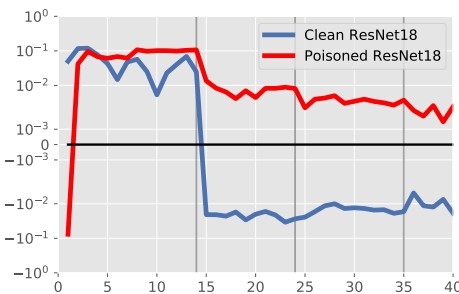

Figure 2: Average batch cosine similarity, per epoch, between the adversarial gradient $\nabla \mathcal{L}_{adv}(\theta)$ and the gradient of each mini-batch $\nabla \mathcal{L}(\theta)$ for a poisoned and a clean ResNet-18. Crucially, the gradient alignment is strictly positive.

Table 1: CIFAR-10 ablation. $\varepsilon = 16$, budget is 1%. Differentiable data augmentation is able to replace a large 8-model ensemble, without increasing computational effort.

| Ensemble | Diff. Data Aug. | Victim does data aug. | Poison Accuracy (%($\pm$SE)) |
|---|---|---|---|
| 1 | X | X | 100.00% ($\pm$0.00) |
| 1 | X | ✓ | 32.50% ($\pm$12.27) |
| 8 | X | X | 78.75% ($\pm$11.77) |
| 1 | ✓ | ✓ | 91.25% ($\pm$6.14) |

dataset will increase adversarial loss. However, after the inclusion of poisoned data, the gradient alignment is modified enough to change the prediction for the target.

## 5 EXPERIMENTAL EVALUATION

We evaluate poisoning approaches in each experiment by sampling 10 random poison-target cases. We compute poisons for each and evaluate them on 8 newly initialized victim models (see supp. material Sec. A.1 for details of our methodology). We refer to only the correct classification of each target as its adversarial class as a success and report as *avg. poison success* the average success rate over all 10 cases, each including 8 poisoned models. We apply algorithm 1 with the following hyperparameters for all our experiments: $\tau = 0.1$, $R = 8$, $M = 250$. We train victim models in a realistic setting, considering data augmentation, SGD with momentum, weight decay and learning rate drops. Code for all experiments can be found at `https://github.com/JonasGeiping/poisoning-gradient-matching`.

### 5.1 EVALUATIONS ON CIFAR-10

As a baseline on CIFAR-10, the inset figure (right) visualizes the number of restarts $R$ and the number of ensembled models $K$, showing that the proposed method is successful in creating poisons even with just a single model (instead of an ensemble). The inset figure shows poison success versus time necessary to compute the poisoned dataset for a budget of 1%, $\varepsilon = 16$ on CIFAR-10 for a ResNet-18. We find that as the number of ensemble models, $K$, increases, it is beneficial to increase the number of restarts as well, but increasing the number of restarts independently also improves perfor-

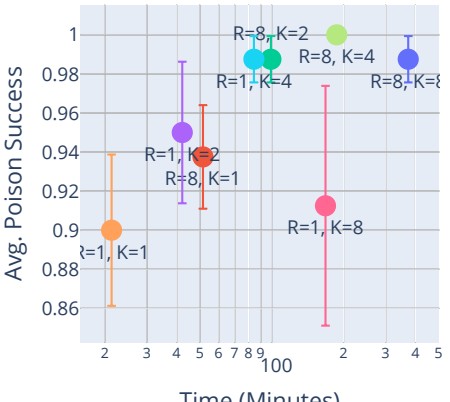

mance. We validate the differentiable data augmentation discussed in section 3.4 in table 1, finding it crucial for scalable data poisoning, being as efficient as a large model ensemble in facilitating robustness.

Next, to test different poisoning methods, we fix our "brewing" framework of efficient data poisoning, with only a single network and diff. data augmentation. We evaluate the discussed gradient matching cost function, replacing it with either the feature-collision objective of Poison Frogs or the bullseye objective of Aghakhani et al. (2020), thereby effectively replicating their methods, but in our context of from-scratch training.

The results of this comparison are collated in table 2. While Poison Frogs and Bullseye succeeded in finetuning settings, we find that their feature collision objectives are only successful in the shallower network in the from-scratch setting. Gradient matching further outperforms MetaPoison on CIFAR-10, while faster (see appendix), in particular as $K = 24$ for MetaPoison.

**Benchmark results on CIFAR-10:** To evaluate our results against a wider range of poison attacks, we consider the recent benchmark proposed in Schwarzschild et al. (2020) in table 3. In the category "Training From Scratch", this benchmark evaluates poisoned CIFAR-10 datasets with a budget of 1% and $\varepsilon = 8$ against various model architectures, averaged over 100 fixed scenarios. We find that the discussed gradient matching attack, even for $K = 1$ is significantly more potent in the more difficult

Table 2: CIFAR-10 Comparison to other poisoning objectives with a budget of $1\%$ within our framework (columns 1 to 3), for a 6-layer ConvNet and an 18-layer ResNet. MetaPoison* denotes the full framework of Huang et al. (2020). Each cell shows the avg. poison success and its standard error.

|  | Proposed | Bullseye | Poison Frogs | MetaPoison* |
|---|---|---|---|---|
| **ConvNet** ($\varepsilon = 32$) | *86.25%* ($\pm 9.43$) | 78.75% ($\pm 7.66$) | 52.50% ($\pm 12.85$) | 35.00% ($\pm 11.01$) |
| **ResNet-18** ($\varepsilon = 16$) | *90.00%* ($\pm 3.87$) | 3.75% ($\pm 3.56$) | 1.25% ($\pm 1.19$) | 42.50 % ($\pm 8.33$) |

Table 3: Results on the benchmark of Schwarzschild et al. (2020). Avg. accuracy of poisoned CIFAR-10 (budget $1\%$, $\varepsilon = 8$) over 100 trials is shown. (*) denotes rows replicated from Schwarzschild et al. (2020). Poisons are created with a ResNet-18 except for the last row, where the ensemble consists of two models of each architecture.

| Attack | ResNet-18 | MobileNet-V2 | VGG11 | Average |
|---|---|---|---|---|
| Poison Frogs* (Shafahi et al., 2018) | 0% | 1% | 3% | 1.33% |
| Convex Polytopes* (Zhu et al., 2019) | 0% | 1% | 1% | 0.67% |
| Clean-Label Backd.* (Turner et al., 2018) | 0% | 1% | 2% | 1.00% |
| Hidden-Trigger Backd.* (Saha et al., 2019) | 0% | 4% | 1% | 2.67% |
| Proposed Attack ($K = 1$) | 45% | 36% | 8% | 29.67% |
| Proposed Attack ($K = 4$) | **55**% | 37% | 7% | 33.00% |
| Proposed Attack ($K = 6$, Heterogeneous) | 49% | **38**% | **35**% | **40.67**% |

benchmark setting. An additional feature of the benchmark is *transferability*. Poisons are created using a ResNet-18 model, but evaluated also on two other architectures. We find that the proposed attack transfers to the similar MobileNet-V2 architecture, but not as well to VGG11. However, we also show that this advantage can be easily circumvented by using an ensemble of different models as in Zhu et al. (2019). If we use an ensemble of $K = 6$, consisting of 2 ResNet-18, 2 MobileNet-V2 and 2 VGG11 models (last row), then the same poisoned dataset can compromise all models and generalize across architectures.

## 5.2 POISONING IMAGENET MODELS

The ILSVRC2012 challenge, "ImageNet", consists of over 1 million training examples, making it infeasible for most actors to train large model ensembles or run extensive hyperparameter optimizations. However, as the new gradient matching attack requires only a single sample of pretrained parameters $\theta$, and operates only on the poisoned subset, it can poison ImageNet images using publicly available pretrained models without ever training an ImageNet classifier. Poisoning ImageNet with previous methods would be infeasible. For example, following the calculations in section 2, it would take over $500$ GPU days (relative to our hardware) to create a poisoned ImageNet for a ResNet-18 via MetaPoison. In contrast, the new attack can poison ImageNet in less than four GPU hours.

Figure 3 shows that a standard ImageNet models trained from scratch on a poisoned dataset "brewed" with the discussed attack, are reliably compromised - with examples of successful poisons shown (left). We first study the effect of varying poison budgets, and $\varepsilon$-bounds (top right). Even at a budget of $0.05\%$ and $\varepsilon$-bound of 8, the attack poisons a randomly initialized ResNet-18 $80\%$ of the time. These results extend to other popular models, such as MobileNet-v2 and ResNet50 (bottom right).
**Poisoning Cloud AutoML:** To verify that the discussed attack can compromise models in practically relevant *black-box setting*, we test against Google's Cloud AutoML. This is a cloud framework that provides access to black-box ML models based on an uploaded dataset. In Huang et al. (2020) Cloud AutoML was shown to be vulnerable for CIFAR-10. We upload a poisoned ImageNet dataset (base: ResNet18, budget $0.1\%$, $\varepsilon = 32$) for our first poison-target test case and upload the dataset. Even in this scenario, the attack is measurably effective, moving the adversarial label into the top-5 predictions of the model in 5 out of 5 runs, and the top-1 prediction in 1 out of 5 runs.

## 5.3 DEFICIENCIES OF DEFENSE STRATEGIES

Previous defenses against data poisoning (Steinhardt et al., 2017; Paudice et al., 2018; Peri et al., 2019) have relied mainly on data sanitization, i.e. trying to find and remove poisons by outlier

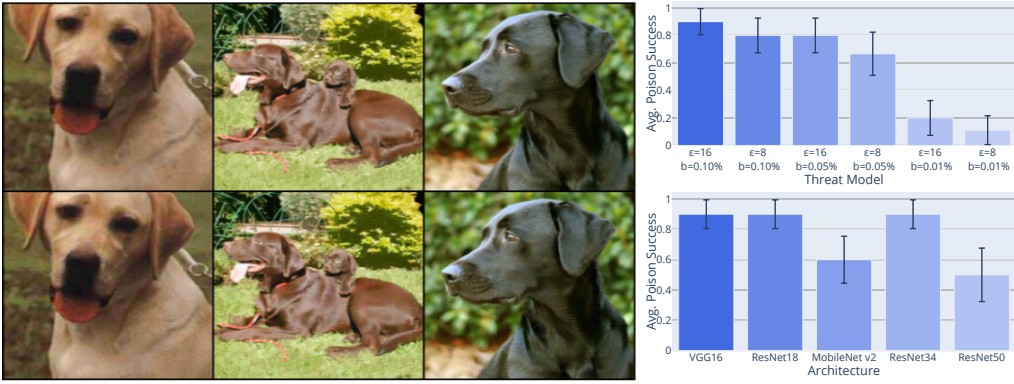

Figure 3: Poisoning ImageNet. **Left**: Clean images (above), with their poisoned counterparts (below) from a successful poisoning of a randomly initialized ResNet-18 trained on ImageNet for a poison budget of $0.1\%$ and an $\ell_\infty$ bound of $\varepsilon = 8$. **Right Top**: ResNet-18 results for different budgets and varying $\varepsilon$-bounds. **Right Bot.**: More architectures (Simonyan & Zisserman, 2014; He et al., 2015; Sandler et al., 2018) with a budget of $0.1\%$ and $\varepsilon = 16$.

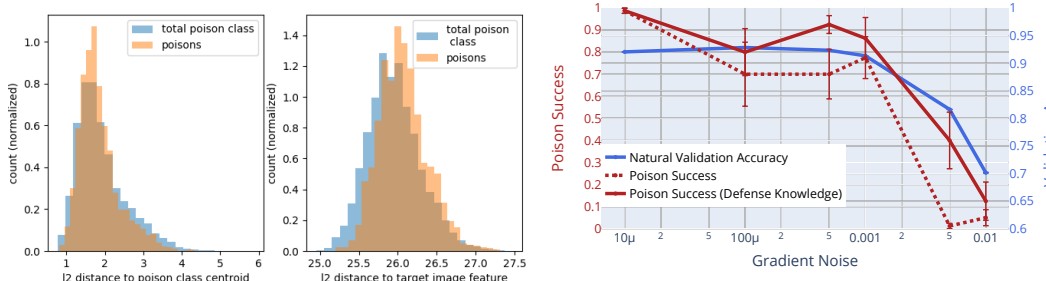

(a) Feature space distance to base class centroid, and target image feature, for victim model on CIFAR-10. $4.0\%$ budget, $\varepsilon = 16$, showing sanitization defenses failing and no feature collision as in Poison Frogs.

(b) Defending through differential privacy. CIFAR-10, $1\%$ budget, $\varepsilon = 16$, ResNet-18. Differential privacy is only able to limit the success of poisoning via trade-off with significant drops in accuracy.

Figure 4: Defense strategies against poisoning.

detection (often in feature space). We demonstrate why sanitization methods fail in the face of the attack discussed in this work in fig. 4a. Poisoned data points are distributed like clean data points, reducing filtering based methods to almost-random guessing (see supp. material, table 6).

Differentially private training is a different defense. It diminishes the impact of individual training samples, in turn making poisoned data less effective (Ma et al., 2019; Hong et al., 2020). However, this come at a significant cost. Figure 4b shows that to push the Poison Success below $15\%$, one has to sacrifice over $20\%$ validation accuracy, even on CIFAR-10. Training a diff. private ImageNet model is even more challenging. From this aspect, differentially private training can be compared to adversarial training (Madry et al., 2017) against evasion attacks. Both methods can mitigate the effectiveness of an adversarial attack, but only by significantly impeding natural accuracy.

## 6   CONCLUSION

We investigate targeted data poisoning via gradient matching and discover that this mechanism allows for data poisoning attacks against fully retrained models that are unprecedented in scale and effectiveness. We motivate the attack theoretically and empirically, discuss additional mechanisms like differentiable data augmentation and experimentally investigate modern deep neural networks in realistic training scenarios, showing that gradient matching attacks compromise even models trained on ImageNet. We close with discussing the limitations of current defense strategies.

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

ACKNOWLEDGEMENTS

We thank the University of Maryland Institute for Advanced Computer Studies, specifically Matthew Baney and Liam Monahan, for their help with the Center for Machine Learning cluster.

This work was supported by DARPA's GARD, QED4RML, and Young Faculty Award program. Additional support was provided by the National Science Foundation Directory of Mathematical Sciences. GT is supported by ONR grant N0001420WX00239, as well as the DoD HPC Modernization Program. WC is supported by NSF DMS 1738003 and MURI from the Army Research Office - Grant No. W911NF-17-1-0304.

## A REMARKS

*Remark* (Validating the approach in a special case). Inner-product loss functions like eq. (3) work well in other contexts. In Geiping et al. (2020), cosine similarity between image gradients was minimized to uncover training images used in federated learning. If we disable our constraints, setting $\varepsilon = 255$, and consider a single poison image and a single target, then we minimize the problem of recovering image data from a normalized gradient as a special case. In Geiping et al. (2020), it was shown that minimizing this problem can recover the target image. This means that we can indeed return to the motivating case in the unconstrained setting - the optimal choice of poison data is insertion of the target image in an unconstrained setting for one image.

*Remark* (Transfer of gradient alignment). An analysis of how gradient alignment often transfers between different parameters and even between architectures has been conducted, e.g. in Charpiat et al. (2019); Koh & Liang (2017) and Demontis et al. (2019). It was shown in Demontis et al. (2019) that the performance loss when transferring an evasion attack to another model is governed by the gradient alignment of both models. In the same vein, optimizing alignment appears to be a useful metric in the case of data poisoning. Furthermore Hong et al. (2020) note that previous poisoning algorithms might already cause gradient alignment as a side effect, even without explicitly optimizing for it.

*Remark* (Poisoning is a Credible Threat to Deep Neural Networks). It is important to understand the security impacts of using unverified data sources for deep network training. Data poisoning attacks up to this point have been limited in scope. Such attacks focus on limited settings such as poisoning SVMs, attacking transfer learning models, or attacking toy architectures (Biggio et al., 2012; Muñoz-González et al., 2019; Shafahi et al., 2018). We demonstrate that data poisoning poses a threat to large-scale systems as well. The approach discussed in this work pertains only to the classification scenario, as a guinea pig for data poisoning, but applications to a variety of scenarios of practical interest have been considered in the literature, for example spam detectors mis-classifying a spam email as benign, or poisoning a face unlock based mobile security systems.

The central message of the data poisoning literature can be described as follows: From a security perspective, the data that is used to train a machine learning model should be under the same scrutiny as the model itself. These models can only be secure if the entire data processing pipeline is secure. This issue further cannot easily be solved by human supervision (due to the existence of clean-label attacks) or outlier detection (see fig. 4a). Furthermore, targeted poisoning is difficult to detect as validation accuracy is unaffected. As such, data poisoning is best mitigated by fully securing the data pipeline.

So far we have considered data poisoning from the industrial side. From the perspective of a user, or individual under surveillance, however, data poisoning can be a means of securing personal data shared on the internet, making it unusable for automated ML systems. For this setting, we especially refer to an interesting application study in Shan et al. (2020) in the context of facial recognition.

## B  EXPERIMENTAL SETUP

This appendix section details our experimental setup for replication purposes. A central question in the context of evaluating data poisoning methods is how to judge and evaluate "average" performance. Poisoning is in general volatile with respect to poison-target class pair, and to the specific target example, with some combinations and target images being in general easier to poison than others. However, evaluating all possible combinations is infeasible for all but the simplest datasets, given that poisoned data has to created for each example and then a neural network has to be trained from scratch every time. Previous works (Shafahi et al., 2018; Zhu et al., 2019) have considered select target pairs, e.g. "birds-dogs" and "airplanes-frogs", but this runs the risk of mis-estimating the overall success rates. Another source of variability arises, especially in the from-scratch setting: Due to both the randomness of the initialization of the neural network, the randomness of the order in which images are drawn during mini-batch SGD, and the randomness of data augmentations, a fixed poisoned dataset might only be effective some of the time, when evaluating it multiple times.

In light of this discussion, we adopt the following methodology: For every experiment we randomly select $n$ (usually 10 in our case) settings consisting of a random target class, random poison class, a random target and random images to be poisoned. For each of these experiments we create a single poisoned dataset by the discussed or a comparing method within limits of the given threat model and then evaluate the poisoned datasets $m$ times (8 for CIFAR-10 and 1 for ImageNet) on random re-initializations of the considered architecture. To reduce randomness for a fair comparison between different runs of this setup, we fix the random seeds governing the experiment and rerun different threat models or methods with the same random seeds. We have used CIFAR-10 with random seeds 1000000000-1111111111 hyperparameter tuning and now evaluate on random seeds 2000000000-2111111111 for CIFAR-10 experiments and 1000000000-1111111111 for ImageNet, with class pairs and target image IDs for reproduction given in tables 4 and 5. For CIFAR-10, the target ID refers to the canonical order of all images in the dataset ( as downloaded from `https://www.cs.toronto.edu/~kriz/cifar.html`); for ImageNet, the ID refers to an order of ImageNet images where the syn-sets are ordered by their increasing numerical value (as is the default in `torchvision`). However for future research we encourage the sampling of new target-poison pairs to prevent overfitting, ideally even in larger numbers given enough compute power.

For every measurement of *avg. poison success* in the paper, we measure in the following way: After retraining the given deep neural network to completion, we measure if the target image is successfully classified by the network as its adversarial class. We do not count mere misclassification of the original label (but note that this usually happens even before the target is incorrectly classified by the adversarial class). Over the $m$ validation runs we repeat this measurement of target classification success and then compute the average success rate for a single example. We then aggregate this average over our 10 chosen random experiments and report the mean and standard error of these average success rates as *avg. poison success*. All error bars in the paper refer to standard error of these measurements.

### B.1  HARDWARE

We use a heterogeneous mixture of hardware for our experiments. CIFAR-10, and a majority of the ImageNet experiments, were run on NVIDIA GEFORCE RTX 2080 Ti gpus. CIFAR-10 experiments

Table 4: Target/poison class pairs generated from the initial random seeds for ImageNet experiments. Target ID relative to CIFAR-10 validation dataset.

| Target Class | Poison Class | Target ID | Random Seed |
|---|---|---|---|
| dog | frog | 8745 | 2000000000 |
| frog | truck | 1565 | 2100000000 |
| frog | bird | 2138 | 2110000000 |
| airplane | dog | 5036 | 2111000000 |
| airplane | ship | 1183 | 2111100000 |
| cat | airplane | 7352 | 2111110000 |
| automobile | frog | 3544 | 2111111000 |
| truck | cat | 3676 | 2111111100 |
| automobile | ship | 9882 | 2111111110 |
| automobile | cat | 3028 | 2111111111 |

Table 5: Target/poison class pairs generated from the initial random seeds for ImageNet experiments. Target Id relative to ILSVRC2012 validation dataset Russakovsky et al. (2015)

| Target Class | Poison Class | Target ID | Random Seed |
|---|---|---|---|
| otter | Labrador retriever | 18047 | 1000000000 |
| warthog | bib | 17181 | 1100000000 |
| orange | radiator | 37530 | 1110000000 |
| theater curtain | maillot | 42720 | 1111000000 |
| hartebeest | capuchin | 17580 | 1111100000 |
| burrito | plunger | 48273 | 1111110000 |
| jackfruit | spider web | 47776 | 1111111000 |
| king snake | hyena | 2810 | 1111111100 |
| flat-coated retriever | alp | 10281 | 1111111110 |
| window screen | hard disc | 45236 | 1111111111 |

were run on $1$ gpu, while ImageNet experiments were run on $4$ gpus. We also use NVIDIA Tesla P100 gpus for some ImageNet experiments. All timed experiments were run using 2080 Ti gpus.

## B.2 MODELS

For our experiments on CIFAR-10 in section 5 we consider two models. In table 2, the "6-layer ConvNet", - in close association with similar models used in Finn et al. (2017) or Krizhevsky et al. (2012), we consider an architecture of 5 convolutional layers (with kernel size 3), followed by a linear layer. All convolutional layers are followed by a ReLU activation. The last two convolutional layers are followed by max pooling with size 3. The output widths of these layers are given by $64, 128, 128, 256, 256, 2304$. In tables 1, 2, in the inset figure and Fig. 4 we consider a ResNet-18 model. We make the customary changes to the model architecture for CIFAR-10, replacing the stem of the original model (which requires ImageNet-sized images) by a convolutional layer of size 3, following by batch normalization and a ReLU. This is effectively equal to upsampling the CIFAR-10 images before feeding them into the model. For experiments on ImageNet, we consider ResNet-18, ResNet-34 (He et al., 2015), MobileNet-v2 (Sandler et al., 2018) and VGG-16 (Simonyan & Zisserman, 2014) in standard configuration.

We train the ConvNet, MobileNet-v2 and VGG-16 with initial learning rate of $0.01$ and the residual architectures with initial learning rate $0.1$. We train for 40 epochs, dropping the learning rate by a factor of 10 at epochs 14, 24, 35. We train with stochastic mini-batch gradient descent with Nesterov momentum, with batch size $128$ and momentum $0.9$. Note that the dataset is shuffled in each epoch, so that where poisoned images appear in mini-batches is random and not known to the attacker. We add weight decay with parameter $5 \times 10^{-4}$. For CIFAR-10 we add data augmentations using horizontal flipping with probability $0.5$ and random crops of size $32 \times 32$ with zero-padding of $4$. For ImageNet we resize all images to $256 \times 256$ and crop to the central $224 \times 224$ pixels. We also consider horizontal flipping with probability $0.5$, and data augmentation with random crops of size $224 \times 224$ with zero-padding of $28$.

When evaluating ImageNet poisoning from-scratch we use the described procedure. To create our poisoned datasets as detailed in Alg. 1, we download the respective pretrained model from `torchvision`, see `https://pytorch.org/docs/stable/torchvision/models.html`.

### B.3 CLOUD AUTOML SETUP

For the experiment using Google's cloud autoML, we upload a poisoned ILSVRC2012 dataset into google storage, and then use `https://cloud.google.com/vision/automl/` to train a classification model. Due to autoML limitations to 1 million images, we only upload up to 950 examples from each class (reaching a training set size slightly smaller than 950 000, which allows for an upload of the 50 000 validation images). We use a ResNet-18 model as surrogate for the black-box learning within autoML, pretrained on the full ILSVRC2012 as before. We create a `MULTICLASS` autoML dataset and specify the vision model to be `mobile-high-accuracy-1` which we train to 10 000 milli-node hours, five times. After training the model, we evaluate its performance on the validation set and target image. The trained models all reach a $69\%$ clean top-1 accuracy on the ILSVRC2012 validation set.

## C  PROOF OF PROPOSITION 1

*Proof of Prop. 1.* Consider the gradient descent update

$$\theta^{k+1} = \theta^k - \alpha_k \nabla \mathcal{L}(\theta^k)$$

Firstly, due to Lipschitz smoothness of the gradient of the adversarial loss $\mathcal{L}_{\text{adv}}$ we can estimate the value at $\theta^{k+1}$ by the descent lemma

$$\mathcal{L}_{\text{adv}}(\theta^{k+1}) \leq \mathcal{L}_{\text{adv}}(\theta^k) - \langle \alpha_k \nabla \mathcal{L}_{\text{adv}}(\theta^k), \nabla \mathcal{L}(\theta^k) \rangle + \alpha_k^2 L ||\nabla \mathcal{L}(\theta^k)||^2$$

If we further use the cosine identity:

$$\langle \nabla \mathcal{L}_{\text{adv}}(\theta^k), \nabla \mathcal{L}(\theta^k) \rangle = ||\nabla \mathcal{L}(\theta^k)|| ||\nabla \mathcal{L}_{\text{adv}}(\theta^k)|| \cos(\gamma^k),$$

denoting the angle between both vectors by $\gamma^k$, we find that

$$\mathcal{L}_{\text{adv}}(\theta^{k+1}) \leq \mathcal{L}_{\text{adv}}(\theta^k) - ||\nabla \mathcal{L}(\theta^k)|| ||\nabla \mathcal{L}_{\text{adv}}(\theta^k)|| \cos(\gamma^k) + \alpha_k^2 L ||\nabla \mathcal{L}(\theta^k)||^2$$

$$= \mathcal{L}_{\text{adv}}(\theta^k) - \left( \alpha_k \frac{||\nabla \mathcal{L}_{\text{adv}}(\theta^k)||}{||\nabla \mathcal{L}(\theta^k)||} \cos(\gamma^k) - \alpha_k^2 L \right) ||\nabla \mathcal{L}(\theta^k)||^2$$

As such, the adversarial loss decreases for nonzero step sizes if

$$\frac{||\nabla \mathcal{L}_{\text{adv}}(\theta^k)||}{||\nabla \mathcal{L}(\theta^k)||} \cos(\gamma^k) > \alpha_k L$$

i.e.

$$\alpha_k L \leq \frac{||\nabla \mathcal{L}_{\text{adv}}(\theta^k)||}{||\nabla \mathcal{L}(\theta^k)||} \frac{\cos(\gamma^k)}{c}$$

for some $1 < c < \infty$. This follows from our assumption on the parameter $\beta$ in the statement of the proposition. Reinserting this estimate into the descent inequality reveals that

$$\mathcal{L}_{\text{adv}}(\theta^{k+1}) < \mathcal{L}_{\text{adv}}(\theta^k) - ||\nabla \mathcal{L}_{\text{adv}}||^2 \frac{\cos(\gamma^k)}{c' L},$$

for $\frac{1}{c'} = \frac{1}{c} - \frac{1}{c^2}$. Due to monotonicity we may sum over all descent inequalities, yielding

$$\mathcal{L}_{\text{adv}}(\theta^0) - \mathcal{L}_{\text{adv}}(\theta^{k+1}) \geq \frac{1}{c' L} \sum_{j=0}^{k} ||\nabla \mathcal{L}_{\text{adv}}(\theta^j)||^2 \cos(\gamma^j)$$

As $\mathcal{L}_{\text{adv}}$ is bounded below, we may consider the limit of $k \to \infty$ to find

$$\sum_{j=0}^{\infty} ||\nabla \mathcal{L}_{\text{adv}}(\theta^j)||^2 \cos(\gamma^j) < \infty.$$

If for all, except finitely many iterates the angle between adversarial and training gradient is less than $90°$, i.e. $\cos(\gamma^k)$ is bounded below by some fixed $\epsilon > 0$, as assumed, then the convergence to a stationary point follows:

$$\lim_{k \to \infty} ||\nabla \mathcal{L}_{\text{adv}}(\theta^k)|| \to 0$$

$\square$

In fig. 5 we visualize measurements of the computed bound from an actual poisoned training. The classical gradient descent converges only if $\alpha_k L < 1$, so we can find an upper bound to this value by 1, even if the actual Lipschitz constant of the neural network training objective is not known to us.

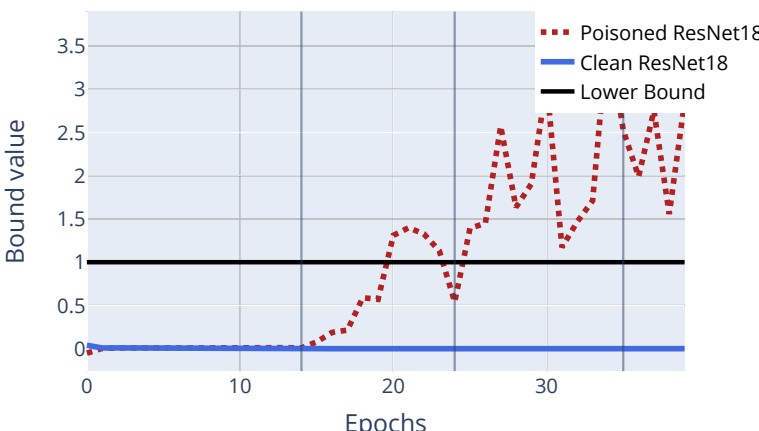

Figure 5: The bound considered in Prop. 1, evaluated during training of a poisoned and a clean model, using a practical estimation of the lower bound via $\alpha_k L \approx 1$. This is an upper bound of $\alpha_k L$ as $\alpha_k < \frac{1}{L}$ is necessary for the convergence of (clean) gradient descent.

## D  POISONED DATASETS

We provide access to poisoned datasets as part of the supplementary material, allowing for a replication of the attack. To save space however, we provide only the subset of poisoned images and not the full dataset. We hope that this separation also aids in the development of defensive strategies. To train a model using these poisoned data points, you can use our code (using `-save full`) our your own to export either CIFAR-10 or ImageNet into an image folder structure, where the clean images can then be replaced by poisoned images according to their ID. Note that the given IDs refer to the dataset ordering as discussed above.

## E  VISUALIZATIONS

We visualize poisoned sample from our ImageNet runs in figs. 6 and 7, noting especially the "clean label" effect. Poisoned data is only barely distinguishable from clean data, even in the given setting where the clean data is shown to the observer. In a realistic setting, this is significantly harder. A subset of poisoned images used to poison Cloud autoML with $\varepsilon = 32$ can be found in fig. 8.

We concentrate only on small $\ell^\infty$ perturbations to the training data as this is the most common setting for adversarial attacks. However, there exist other choices for attacks in practical settings. Previous works have already considered additional color transformations (Huang et al., 2020) or watermarks (Shafahi et al., 2018). Most techniques that create adversarial attacks at test time within various

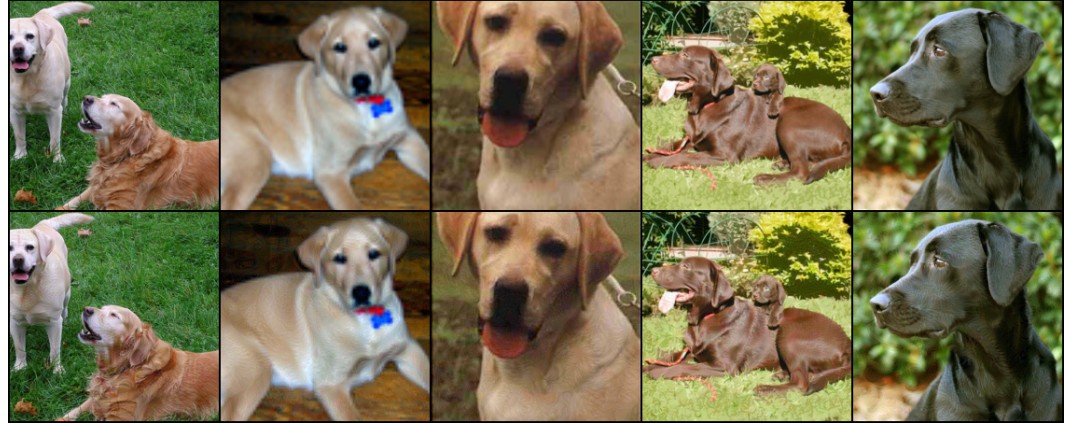

Figure 6: Clean images (above), with their poisoned counterparts (below) from a successful poisoning of a ResNet-18 model trained on ImageNet. The poisoned images (taken from the Labrador Retriever class) successfully caused mis-classification of a target (otter) image under a threat model given by a budget $0.1\%$ and an $\ell_\infty$ bound of $\varepsilon = 8$.

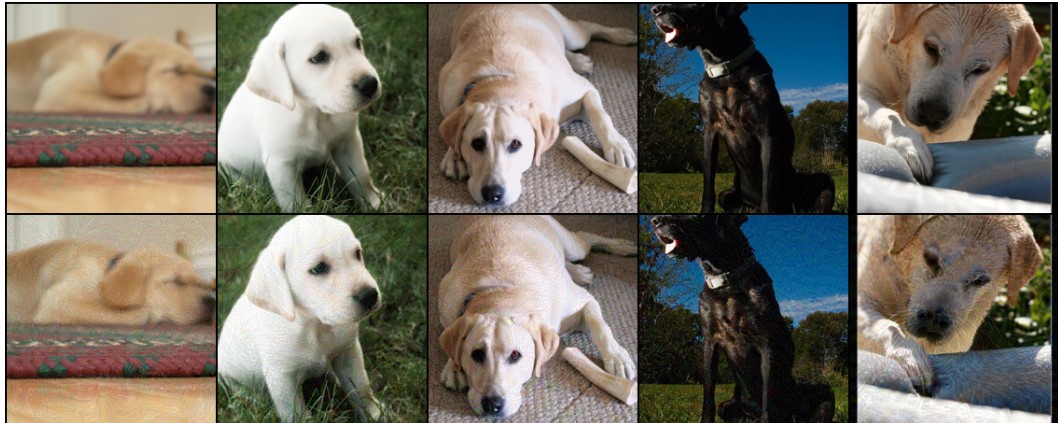

Figure 7: Clean images (above), with their poisoned counterparts (below) from a successful poisoning of a randomly initialized ResNet-18 trained on ImageNet. The poisoned images (taken from the Labrador Retriever class) successfully caused mis-classification of a target (otter) image under a threat model given by a budget of $0.1\%$ and an $\ell_\infty$ bound of $\epsilon = 16$.

constraints (Engstrom et al., 2017; Zhang et al., 2019; Heng et al., 2018; Karmon et al., 2018) are likely to transfer into the data poisoning setting. Likewise, we do not consider hiding poisoned images further by minimizing perceptual scores and relate to the large literature of adversarial attacks that evade detection (Carlini & Wagner, 2017).

In fig. 9 we visualize how the adversarial loss and accuracy behave during an exemplary training run, comparing the adversarial label with the original label of the target image.

# F  ADDITIONAL EXPERIMENTS

This section contains additional experiments.

## F.1  FULL-SCALE METAPOISON COMPARISONS ON CIFAR-10

Removing all constraints for time and memory, we visualize time/accuracy of our approach against other poisoning approaches in fig. 10. Note that attacks, like MetaPoison, which succeed on CIFAR-10 only after removing these constraints, cannot be used on ImageNet-sized datasets due to the

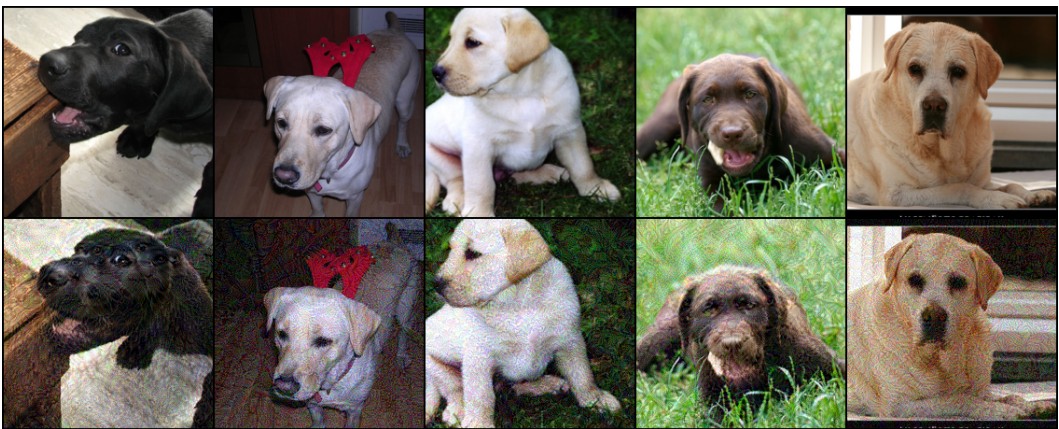

Figure 8: Clean images (above), with their poisoned counterparts (below) from a successful poisoning of a Google Cloud AutoML model trained on ImageNet. The poisoned images (taken from the Labrador Retriever class) successfully caused mis-classification of a target (otter) image. This is accomplished with a poison budget of $0.1\%$ and an $\ell_\infty$ bound of $\varepsilon = 32$ - the black-box attack against autoML requires an increased perturbation magnitude, in contrast to the other gray-box experiments in this work.

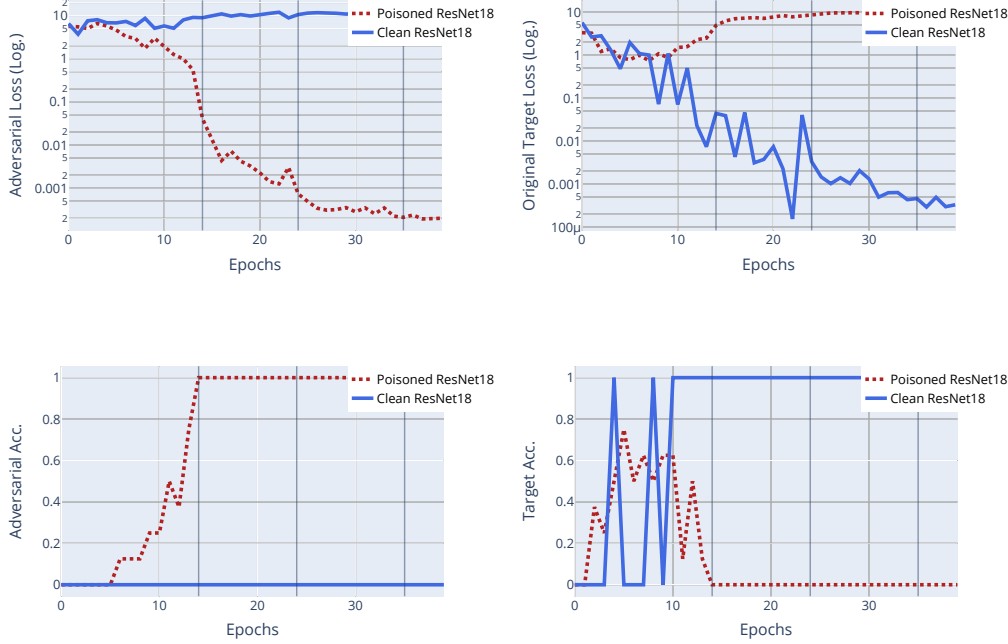

Figure 9: Cross entropy loss (Top) and accuracy (Bottom) for a given target with its adversarial label (left), and with its original label (right) shown for a poisoned and a clean ResNet-18. The clean model is used as victim for the poisoned model. The loss is averaged 8 times for the poisoned model. Learning rate drops are marked with gray horizontal bars.

significant computational effort required. For MetaPoison, we use the original implementation of Huang et al. (2020), but add our larger models. We find that with the larger architectures and different threat model (original MetaPoison considers a color perturbation in addition to the $\ell_\infty$ bound), our gradient matching technique still significantly outperforms MetaPoison. Note that for the ConvNet experiment on MetaPoison in table 2, we found that MetaPoison seems to overfit with $\varepsilon = 32$, and as

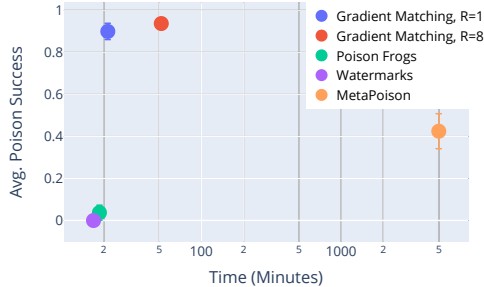

Figure 10: CIFAR-10 comparison without time and memory constraints for a ResNet18 with realistic training. Budget $1\%$, $\varepsilon = 16$. Note that the x-axis is logarithmic.

such we show numbers running the MetaPoison code with $\varepsilon = 16$ in that column, which are about $8\%$ better than $\varepsilon = 16$. This is possibly a hyperparameter question for MetaPoison, which was optimized for $\varepsilon = 8$ and a color perturbation.

## F.2 DEFICIENCIES OF FILTERING DEFENSES

Defenses aim to sanitize training data of poisons by detecting outliers (often in feature space), and removing or relabeling these points (Steinhardt et al., 2017; Paudice et al., 2018; Peri et al., 2019). In some cases, these defenses are in the setting of general performance degrading attacks, while others deal with targeted attacks. By in large, poison defenses up to this point are limited in scope. For example, many defenses that have been proposed are specific to simple models like linear classifiers and SVM, or the defenses are tailored to weaker attacks such as collision based attacks where feature space is well understood (Steinhardt et al., 2017; Paudice et al., 2018; Peri et al., 2019). However, data sanitization defenses break when faced with stronger attacks. Table 6 shows a defense by anomaly filtering. averaged over 6 randomly seeded poisoning runs on CIFAR-10 (4% budget w/ $\varepsilon = 16$), we find that outlier detection is only marginally more successful than random guessing.

Table 6: Outlier detection is close to random-guessing for poison detection on CIFAR-10.

|  | 10% filtering | 20% filtering |
| --- | --- | --- |
| Expected poisons removed (outlier method) | 248 | 467 |
| Expected clean removed (outlier method) | 252 | 533 |
| Expected poisons removed (random guessing) | 200 | 400 |
| Expected clean removed (random guessing) | 300 | 600 |

## F.3 DETAILS: DEFENSE BY DIFFERENTIAL PRIVACY

In fig. 4b we consider a defense by differential privacy. According to Hong et al. (2020), gradient noise is the key factor that makes differentially private SGD (Abadi et al., 2016) useful as a defense. As such we keep the gradient clipping fixed to a value of 1 and only increase the gradient noise in fig. 4b. To scale differentially private SGD, we only consider this gradient clipping on the mini-batch level, not the example level. This is reflected in the red, dashed line. A trivial counter-measure against this defense is shown as the solid red line. If the level of gradient noise is known to the attacker, then the attacker can brew poisoned data by the approach shown in algorithm 1, but also add gradient noise and gradient clipping to the poison gradient. We use a naive strategy of redrawing the added noise every time the matching objective $\mathcal{B}(\Delta, \theta)$ is evaluated. It turns out that this yields a good baseline counter-attack against the defense through differential privacy.

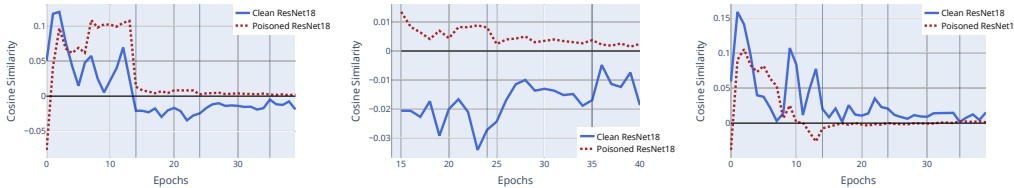

(a) Alignment of $\nabla\mathcal{L}_{\text{adv}}(\theta)$ and $\nabla\mathcal{L}(\theta)$

(b) Zoom: Alignment of $\nabla\mathcal{L}_{\text{adv}}(\theta)$ and $\nabla\mathcal{L}(\theta)$ from epoch 14.

(c) Alignment of $\nabla\mathcal{L}_{\text{t}}(\theta)$ (orig. label) and $\nabla\mathcal{L}(\theta)$

Figure 11: Average batch cosine similarity, per epoch, between the adversarial gradient and the gradient of each mini-batch (left), and with its clean counterpart $\nabla\mathcal{L}_{\text{t}}(\theta) := \nabla_\theta\mathcal{L}(x^t, y^t)$ (right) for a poisoned and a clean ResNet-18. Each measurement is averaged over an epoch. Learning rate drops are marked with gray vertical bars.

### F.4 Details: Gradient Alignment Visualization

Figure 11 visualizes additional details regarding fig. 2. Figure 11a replicates fig. 2 with linear scaling, whereas fig. 11b shows the behavior after epoch 14, which is the first learning rate drop. Note that in all figures each measurement is averaged over an epoch and the learning rate drops are marked with gray vertical bars. Figure 11c shows the opposite metric, that is the alignment of the original (non-adversarial) gradient. It is important to note for these figures, that the positive alignment is the crucial, whereas the magnitude of alignment is not as important. As this is the gradient averaged over the entire epoch, the contributions are from mini-batches can contain none or only a single poisoned example.

### F.5 Ablation Studies - Reduced Brewing/Victim Training Data

In order to further test the strength and possible limitations of the discussed poisoning method, we perform several ablation studies, where we reduce either the training set known to the attacker or the set of poisons used by the victim, or both.

In many real world poisoning situations, it is not reasonable to assume that the victim will unwittingly add all poison examples to their training set, or that the attacker knows the full victim training set to begin with. For example, if the attacker puts 1000 poisoned images on social media, the victim might only scrape 300 of these. We test how dependent the method is on the victim training set by randomly removing a proportion of data (clean + poisoned) from the victim's training set. We then train the victim on the ablated poisoned dataset, and evaluate the target image to see if it is misclassified by the victim as the attacker's intended class. Then, we add another assumption - the brewing network does not have access to all victim training data when creating the poisons (see tab 7). We see that the attacker can still successfully poison the victim, even after a large portion of the victim's training data is removed, or the attacker does not have access to the full victim training set.

Table 7: Average poisoning success under victim training data ablation. In the first regime, victim ablation, a proportion of the victim's training data (clean + poisoned) is selected randomly and then the victim trains on this subset. In the second regime, pretrained + victim ablation, the pretrained network is trained on a randomly selected proportion of the data, and then the victim chose a new random subset of clean + poisoned data on which to train. All results averaged over 5 runs on ImageNet.

|  | 70% data removed | 50% data removed |
|---|---|---|
| victim ablation | 60% | 100% |
| pretrained + victim ablation | 60% | 80% |

### F.6 Ablation Studies - Method

Table 8 shows different variations of the proposed method. While using the Carlini-Wagner loss as a surrogate for cross entropy helped in Huang et al. (2020), it does not help in our setting. We

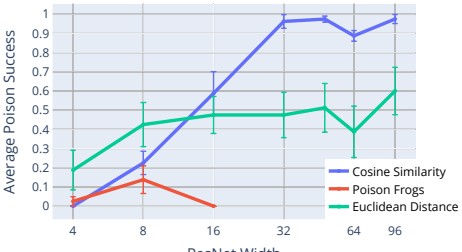 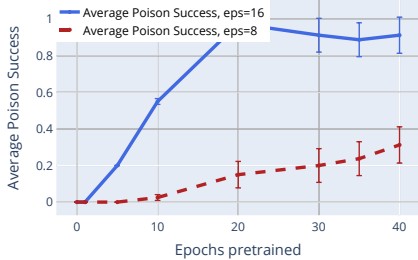

Figure 12: Ablation Studies. Left: avg. poison success for Euclidean Loss, cosine similarity and the Poison Frogs objective (Shafahi et al., 2018) for thin ResNet-18 variants. Right: Avg. poison success vs number of pretraining epochs.

Table 8: CIFAR-10 ablation runs. $\varepsilon = 16$, budget is $1\%$. All values are computed for ResNet-18 models.

| Setup | Avg. Poison Success %($\pm$SE) | Validation Acc.% |
|---|---|---|
| Baseline (full data aug., $R = 8$, $M = 250$ | 91.25% ($\pm$6.14) | 92.20% |
| Carlini-Wagner loss instead of $\mathcal{L}$ | 77.50% ($\pm$9.32) | 92.08% |
| Fewer Opt. Steps ($M = 50$) | 40.00% ($\pm$10.87) | 92.05% |
| Euclidean Loss instead of cosine sim. | 61.25% ($\pm$9.75) | 92.09% |

further find that running the proposed method for only 50 steps (instead of 250 as everywhere else in the paper) leads to a significant loss in avg. poison success. Lastly we investigate whether using euclidean loss instead of cosine similarity would be beneficial. This would basically imply trying to match eq. (2) directly. Euclidean loss amounts to removing the invariance to gradient magnitude, in comparison to cosine similarity, which is invariant. We find that this is not beneficial in our experiments, and that the invariance with respect to gradient magnitude does allow for the construction of stronger poisoned datasets. Interestingly the discrepancy between both loss functions is related to the width of the network. In fig. 12 on the left, we visualize avg. poison success for modified ResNet-18s. The usual base width of 64 is replaced by the width value shown on the x-axis. For widths smaller than 16, the Euclidean loss dominates, but its effectiveness does not increase with width. In contrast the cosine similarity is superior for larger widths and seems to be able to make use of the greater representative power of the wider networks to find vulnerabilities. fig. 12 on the right examines the impact of the pretrained model that is supplied to algorithm 1. We compare avg. poison success against the number of pretraining epochs for a budget of $1\%$, first with $\varepsilon = 16$ and then with $\varepsilon = 8$. It turns out that for the easier threat model of $\varepsilon = 8$, even pretraining to only 20 epochs can be enough for the algorithm to work well, whereas in the more difficult scenario of $\varepsilon = 8$, performance increases with pretraining effort.

### F.7 TRANSFER EXPERIMENTS

In addition to the fully black-box pipeline of the AutoML experiments in appendix B, we test the transferability of our poisoning method against other commonly used architectures. Transfer results on CIFAR-10 can be found in table 3. On Imagenet, we brew poisons with a variety of networks, and test against other networks. We find that poisons crafted with one architecture can transfer and cause targeted mis-classification in other networks (see fig. 13).

### F.8 MULTI-TARGET EXPERIMENTS

We also perform limited tests on poisoning multiple targets simultaneously. We find that while keeping the small poison budget of $1\%$ fixed, we are able to successfully poison more than one target while optimizing poisons simultaneously, see table 9. Effectively, however, every target image gradient has to be matched with an increasingly smaller budget. As the target images are drawn at

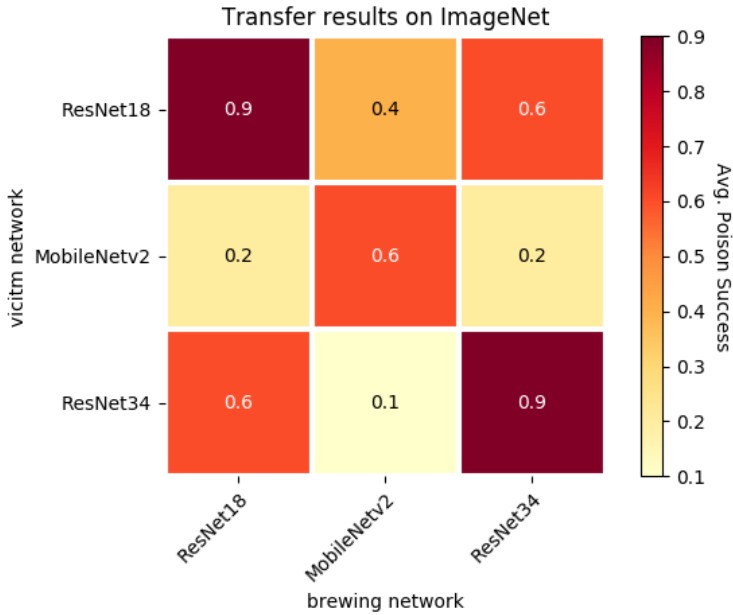

Figure 13: Direct transfer results on common architectures. Averaged over 10 runs with budget of 0.1% and $\varepsilon$-bound of 16. Note that for these transfer experiments, the model was *only* trained on the "brewing" network, without knowledge of the victim. This shows a transferability to unknown architectures.

Table 9: CIFAR-10 ablation runs. $\varepsilon = 16$, budget is 1% with default parameters, modifying only the number of targets and budget.

| Targets | Budget | Avg. Poison Success | Budget / Target | Success × Target |
|---|---|---|---|---|
| 1 | 1% | 90.00% (±4.74) | 1% | 90 |
| 1 | 4% | 95.00% (±4.87) | 4% | 95 |
| 1 | 6% | 90.00% (±6.71) | 6% | 90 |
| 2 | 1% | 65.00% (±7.16) | 0.5% | 130 |
| 4 | 1% | 36.25% (±4.50) | 0.25% | 145 |
| 4 | 4% | 65.00% (±5.12) | 1% | 260 |
| 4 | 6% | 46.25% (±6.19) | 1.5% | 185 |
| 5 | 5% | 44.00% (±5.40) | 1% | 220 |
| 6 | 6% | 30.83% (±5.92) | 1% | 185 |
| 8 | 4% | 25.62% (±3.59) | 0.5% | 205 |
| 16 | 4% | 18.12% (±2.86) | 0.25% | 290 |

Table 10: CIFAR-10 baseline clean validation accuracy. $\varepsilon = 16$, budget is $1\%$. All values are computed for ResNet-18 models as in the baseline plot in section 5.1.

| Setting | Unpoisoned | Poisoned |
|---|---|---|
| $K = 1, R = 1$ | 92.25% ($\pm 0.10$) | 92.12% ($\pm 0.05$) |
| $K = 2, R = 1$ | 92.16% ($\pm 0.08$) | 92.06% ($\pm 0.04$) |
| $K = 4, R = 1$ | 92.18% ($\pm 0.05$) | 92.08% ($\pm 0.04$) |
| $K = 8, R = 1$ | 92.16% ($\pm 0.04$) | 92.20% ($\pm 0.03$) |
| $K = 1, R = 8$ | 92.22% ($\pm 0.11$) | 92.08% ($\pm 0.04$) |
| $K = 2, R = 8$ | 92.27% ($\pm 0.07$) | 92.03% ($\pm 0.05$) |
| $K = 8, R = 8$ | 92.13% ($\pm 0.05$) | 92.04% ($\pm 0.03$) |

random and not semantically similar (aside from their shared class), their synergy is limited. As such, we generally require a larger budget for multiple targets. We show various combinations of number of targets and budget in table 9. While it is possible to reach near-100% avg. poison success for a single target in the setting considered in this work, this value is not reached when optimizing for multiple targets, even when the budget is increased - although the total number of erroneous classifications increases. We analyze this via the last column in table 9, showing avg. poison success multiplied by number of targets, where we find that multiple targets with increased budget can lead to more total mis-classifications, e.g. a score of 290 for 16 targets and a budget of $4\%$, yet the success for each individual target is only 18% on average.

## F.9    NO IMPACT ON VALIDATION ACCURACY

The discussed attack does not significantly alter clean validation accuracy (i.e. validation accuracy on all validation images besides the targets), as the attack is specifically tailored to align only to specific target gradients, and as only a small budget of images is changed within $\varepsilon$ bounds. We validate this by reporting the clean validation accuracy for the CIFAR-10 baseline experiment in section 5.1 in table 10, finding that a drop in validation accuracy is on the order of $0.1\%$.

