# OpenReview forum: "Witches' Brew: Industrial Scale Data Poisoning via Gradient Matching"
_ICLR.cc/2021/Conference — ICLR 2021 Poster_

### Official Review · AnonReviewer3 · 2020-10-17
**Important empirical work demonstrating real threat of poisoning attack on large-scale CNNs.**

**Rating:** 7
**Confidence:** 4

**Review:**

This paper presents a scalable data poisoning attack algorithm focusing on targeted attacks. The technique is based on gradient matching, where the intuition is to design the poisoning patterns such that their effect on the gradient of the training loss mimics the gradient as if the targeted test image is included in the training data.

The paper presents both theoretical intuitions behind the algorithm, as well as empirical reduction and simplification to make the algorithm scalable to ImageNet and applicable to even a black-box attack against the Google Cloud AutoML toolkit.

The algorithm proposed in this paper is practical and general, making it a realistic poisoning threat to modern deep learning systems. The presentation is clear and the theoretical justification is intuitive and easy to understand.

Overall, I think this paper is a good contribution to the study of the large-scale poisoning attack.

Minor typo:
 In proof of Prop 1, you need the angle between the two gradients to be almost always smaller than 90 degrees, not 180 degrees.

---

> ### Author Response · Authors · 2020-11-13
> **Response to Reviewer 3**
>
> Thank you for your support and sharing our enthusiasm about this work. We will gladly fix the typo in proof 1.

---

### Official Review · AnonReviewer2 · 2020-10-29
**New and practical targeted poisoning attack**

**Rating:** 6
**Confidence:** 4

**Review:**

This paper proposed a simple yet effective approach for data poisoning attack targeting a few "clean-label" victim images, using the idea of gradient matching (cosine similarity maximization) between the gradients of adversarial and clean losses. Although the attack model still requires knowing the network architecture (gray-box setting), the resulting poisoned datasets are more effective against different initializations, and some techniques (e.g. model ensemble, multiple restarts) are proposed to further boost the attack performance. The attack results are significantly better than the compared poisoning attacks, and the authors show effective attacks on the ImageNet dataset as well as Google Cloud AutoML with the poisoned data. The authors also discussed the proposed attack on some defenses, showing that the poison has limited change to feature distribution, and differential privacy can mitigate the attack but at the cost of reduced utility (clean accuracy).

Overall, this paper shows some new insights and sets new benchmarks for targeted data poisoning attacks, with practical threat assessment on ImageNet datasets and Google Cloud AutoML, which I deem as a significant contribution. The proposed gradient matching is simple, intuitive, yet very effective. One limitation from Appendix A.8 is that the proposal may not scale well to more than 1 target image, as indicated by the rapidly decreasing attack accuracy. It will be more meaningful to control the effective budget/target and check the resulting accuracy of different number of targets, in order to understand whether gradient matching is scalable to multiple-target setting.

---

> ### Author Response · Authors · 2020-11-13
> **Answer to Reviewer 2**
>
> **“Although the attack model still requires knowing the network architecture (gray-box setting), the resulting poisoned datasets are more effective against different initializations, and some techniques (e.g. model ensemble, multiple restarts) are proposed to further boost the attack performance.”**
> While the attack is most successful when the attacker has knowledge of the victim’s architecture, we stress that this is not a requirement for a successful attack, as demonstrated in Table 3, with our Google Cloud AutoML results and Fig. 13. In Table 3 and Fig.13 we show that attacks directly transfer to other architectures. We also show that an ensemble of several architectures can attack any of the ensembled architectures and as such the attacker can ensemble common architectures. Lastly, for the google Cloud experiments the architecture is entirely unknown.
>
> **“One limitation from Appendix A.8 is that the proposal may not scale well to more than 1 target image, as indicated by the rapidly decreasing attack accuracy. It will be more meaningful to control the effective budget/target and check the resulting accuracy of different number of targets, in order to understand whether gradient matching is scalable to multiple-target setting.”**
> While you are correct that the attack success decreases (in percentage) as we increase the number of targets, this is for a fixed budget. It would be interesting going forward to test the effect of scaling the budget with the number of targets.

---

> > ### Comment · AnonReviewer2 · 2020-11-17
> > **My review comments are well addresed**
> >
> > I thank the authors for providing additional numerical results and clarifying my questions. I have no further comments. Neat work!

---

### Official Review · AnonReviewer1 · 2020-10-29
**Blind review**

**Rating:** 7
**Confidence:** 3

**Review:**

## Summary
- The paper proposes a novel data poisoning attack i.e., to perturb a small fraction of images in the victim's training dataset so as to cause targeted misclassification on certain examples at inference time.
- The proposed approach works by perturbing the clean poison set to introduce a gradient direction which mimics the victim training their model on a targeted mislabeled set.
- Experiments on CIFAR10 and ImageNet demonstrate that the model outperforms competing approaches.

---

## Strengths

**1. Attack insight**
- I appreciate the insight used to craft the perturbations for the poisoned instances. It seems reasonable to me to exploit the fact that the poisoned instances are used for training using a gradient-descent approach; using the gradient information as a yardstick to craft the perturbations is a nice insight that the paper leverages.

**2. Thorough evaluation**
- I am impressed by the thoroughness in the evaluation. The authors extensively evaluate numerous factors influencing the performances (e.g., size of ensembles, no. of restarts), compare with recent baselines, etc.
- To add to it, the authors further evaluate on Imagenet and achieve strong results.

**3. Writing**
- I enjoyed the writing in the paper. I found the presentation clear and easy to follow.

---

## Concerns

### Major Concerns

**1. Attack assumes access to exact training data**
- If I understand the approach correctly, it assumes access to the exact dataset used by the victim to train the model? Isn't this a really strong assumption?
- Because if this is the case in the threat scenario, couldn't the attacker simply poison the entire dataset?
- As a result, I wonder whether the attack also extends to the more interesting and practical case where the adversary has limited access to the victim's training set.

**2. From scratch**
- At many times in the paper, the authors remark that the attack works in spite of the targeted model being trained from scratch from an unknown initialization.
- However, I would suspect that it is easier to tailor the poisoned instances with access to a strong gradient signal, such as early on during training. Are the authors aware whether the approach is robust to victim models that has been pretrained?

### Minor Concerns

**3. Test accuracies**
- Could the authors comment on the difference in victim's test-set accuracy training with the clean and poisoned training set? I found this largely missing, since the focus primarily seems to be on the accuracy on the target set.
- Because a minor concern I have is that the victim model might be overfitting to the poisoned instances by trading off test-set accuracy. It would be nice to know how severe this is.

**4. "single differentiation"**
- The authors claim that the attack requires only a "single differentiation". But doesn't the model have to be twice differentiable ($\nabla_x \nabla_\theta \mathcal{L}(\cdot)$) to perform the updates?


### Nitpicks

**5. Strong focus on MetaPoison**
- The paper makes many head-to-head comparisons with MetaPoison. I'm not sure why, since MetaPoison doesn't seem too closely-related. Especially in S5.2, it appears that it is singled out to demonstrate the computational overhead. This is understandable since it's relies on a meta-learning approach.

**6. Writing**
- Fig 4b is unreadable -- I recommend resizing the figure.
- It seems surprising that the related work section claims that poisoning attacks, unlike backdoor attacks, do not require access to test data. As I'm aware, both require the same access to test-set -- specifically that a particular test instance is presented at inference time to cause misclassification. In fact I would think backdoors are more generalizable here since any test instance can be watermarked to introduce misclassification, unlike pre-specified instances in the case of poisoning.

### Post-rebuttal update
I thank the authors for their response -- this helps. Having read the other reviews, I am still leaning towards acceptance.

---

> ### Author Response · Authors · 2020-11-13
> **Answer to Reviewer 1**
>
> **1. Attack assumes access to exact training data**
>
> The assumption of knowledge of full training set is a white-box scenario in which the attack is most dangerous. However, we do conduct experiments where the poisons are created on a different dataset than the victim is trained on. We show these in appendix Table 7, where we find that poisons are still effective even if only a subset of CIFAR-10 is known to the attacker. Also note that the proposed attack only needs access to a pretrained model trained on a dataset similar to the victim dataset and access to only  the subset of data that is supposed to be poisoned - only these images are included in the gradient matching objective. Attacks where all data is modified are possible within our framework, this would correspond to a budget of 100%, and likely consider a smaller perturbation - but we did not focus on these attacks, because the attacker might only be able to modify a small subset of data.
>
>  Attacks where all the data is modified are certainly possible, but we focus on the more strict threat model wherein an attacker might have knowledge about what training data a victim will use, but only be able to modify a small portion of this data.
>
>
> **2. From scratch**
> The question is on what data would the victim pretrain? If the dataset is poisoned then even this pretraining would happen using poisoned data. Are you referring to a transfer learning setting like the one discussed in Shafahi et al?
> Or is the question whether the gradients should be aligned based on an early epoch? We include a comparison in Fig 12b, where we analyze the strength of the attack when using a pretrained model that is trained for fewer epochs - we find that using pretrained models from later epochs leads to more successful attacks. Note that although the gradient signal is small in magnitude, the magnitude is cancelled in the cosine similarity. We also analyzed the effects of the poisons at different stages of training, see Fig. 9.. We find that in general, the victim begins to misclassify the target image in the last 20 epochs, leading us to believe the poisoned gradient starts to take hold later in training.
>
> **3. Test accuracies**
> Please refer to our general comment for information about clean test accuracies.
>
> **4. "single differentiation"**
> We will clarify this statement - the statement is in the context of other bilevel methods, which need several evaluations of “the” gradient $ \nabla_x \nabla_\theta \mathcal{L}(\cdot)$ to take a single step - but indeed two backpropagations are necessary to compute an update to the poisoned data.
>
> **5. Strong Focus on Metapoison:**
> We focus on MetaPoison because to our knowledge, this is the only other method that performs targeted, clean-label poisoning from scratch. We do agree though that the methods are quite different in their motivation/approaches.
>
> **6. Writing**
> Thank you for the comments. We made some figures in-line/with double subfigures which may have made the legend hard to read without zooming in. We will expand these figures for the final version, and can post an enlarged version to the appendix in the meantime.
> As for requiring access to the test set, there is a subtle, but important difference in the attacker assumptions. For poisoning attacks like ours, we only require the attacker to have picked out a specific target instance that they wish to have the victim misclassify. We do not assume the attacker can modify this target image, as is the case with backdoor attacks. One could imagine this distinction being important if the attacker wishes to poison a facial recognition system where the target is an unsuspecting third party, the attacker will not be able to add perturbations to the images of this party’s face. Also, if the target is entirely unmodified, then there is no chance for a defender to sanitize the target image at test-time, as is possible for backdoor attacks (see e.g. “Wang et al. Neural Cleanse: Identifying and Mitigating Backdoor Attacks in Neural Networks” - a test-time defense against trigger patches).

---

### Official Review · AnonReviewer4 · 2020-11-05
**Review for Witches' Brew: Industrial Scale Data Poisoning via Gradient Matching**

**Rating:** 5
**Confidence:** 5

**Review:**

Summary: This paper introduces a novel targeted clean-label poisoning attack, expected to be more efficient and scalable than current ones. The attack is formulated as a bilevel problem which is then solved with a (fast) heuristic approach based on aligning the gradients of the inner and outer objective functions. A theoretical analysis is also reported to show that this strategy consistently finds a descent direction for the outer objective, asymptotically converging to (a local) minimum.

First of all, I like how the authors have derived their attack and its heuristic solution, and I'm wondering if this generalizes to other applications where bilevel problems are at the core, like meta learning or hyperparameter optimization. However, I have several concerns on the presentation and soundness of the reported results.

First, I think that this paper makes confusion (at least in the reader's mind) when introducing the data poisoning problem. At the beginning, there is no clear distinction between the two main goals of data poisoning:
(1) poisoning availability attacks, which aim to increase the test error causing a denial of service, and
(2) poisoning integrity attacks (which are often referred to -in a misleading manner- as targeted attacks), which aim to allow specific intrusions/attacks at test time (backdoor attacks belong to this category).
For a clearer nomenclature/definition see: M. Barreno, B. Nelson, A. Joseph, and J. Tygar. The security of machine learning. Machine Learning, 81:121–148, 2010.
There, indeed, targeted/untargeted is referred to the victim user, not to the goal/security violation caused by the attack.

While this work seems to claim that, in general, poisoning attacks are computationally demanding, a distinction should be made. While poisoning availability attacks are typically much more computationally demanding (they do require solving the bilevel optimization problem to work well) and this heavily hinders their scalability to large datasets, poisoning integrity attacks can be quite efficient (there's no need to solve a bilevel problem for them to work well, as in the case of most of the considered competing attacks in this paper, or anyway the approach can be simplified - see, e.g., Koh et al., ICML 2017, where the whole network was frozen and the bilevel problem was solved only by assuming that the parameters of the last layer were updated).
I believe that this aspect should be clarified from the beginning. First, this paper focuses on *targeted* (or integrity) poisoning attacks, and this should also be clear from the title. Second, sentences related to the overwhelming complexity of targeted/integrity poisoning attacks should be revised (e.g., Koh et al., ICML 2017 also worked on the DogFish data which should be a subset of ImageNet, if I'm not mistaken).


Another important issue which I do not completely understand is what the authors mean with the word "from scratch", from the viewpoint of previous attacks. I agree with them that previous attacks designed to work on pre-trained models with fine tuning may not work against models which are trained from scratch, but what prevents the attacker to train a model from scratch on the clean data and then design their attack samples with fine tuning? The attack samples can then be added to the initial training set to see if, when learning again from scratch on the poisoned data, the attack remains effective or not. Is this the setup that the authors have considered in their paper for such attacks, or they run them against an "untrained" (or not fully trained) model? If we are in the second scenario, I don't think the analysis reported should be considered fair enough.


There are parts in the paper where it is claimed that 'clean-label' attacks are in some sense better than label flips or attacks that do not preserve image semantics. Why? Are we expecting human labelers to check the quality of our data?
Or are we expecting that clean-label attacks are harder to spot?
Both questions are unaddressed in this paper.
First, I don't think that in many realistic scenarios humans are expected to cleanup the whole training set, especially when it contains a lot of samples. Second, it's also true that the level of noise used in this paper is not so small. By zooming in Figs 6-8, the perturbation becomes quite visible even to the human eye.
Hence, we cannot only instruct humans to detect these patterns, but we can probably train detectors to do that automatically. Accordingly, I don't see in which practical, relevant application scenarios "clean-label" can be retained useful as a requirement.
Finally, even though the authors have analyzed the robustness of their attack against some defense mechanisms, the defenses considered aim to detect mostly poisoning availability attacks and NOT backdoor attacks or targeted/integrity poisoning.
I am even skeptical that such methods can detect label flips or even other current attacks. Have the authors tested such defenses against the competing approaches (poisoning frogs, convex polytopes, etc.)? Do these attacks work or not against them?
How do detection methods for backdoor attacks work against the proposed attack? For a list of such detection methods, see, e.g., Table 1 in https://arxiv.org/abs/1910.03137 (note that some detection methods should work against clean-label attacks too, there's no need to put a trigger on the image).


Finally, the experimental section is missing key information for reproducing the experiments. The parameters \tau, R and M are given a value but not a definition. The figure with the average accuracy vs. time is missing a caption and a figure label. It is extremely unclear what this figure is showing as 1) the parameters are missing descriptions 2) the metric used for evaluating the figure is described nowhere in the paper. This problem also extends to tables 1, 2 and 3: a clear definition of the "evaluation metric" should be given. Are we interested more in preserving accuracy or in the attack success? How is poisoning success defined? (this might be explained in the supplementary material, but it is important for understanding the whole results). Why not including a plot with poisoning success vs. accuracy of the model?

To summarize, the paper is promising, but important details and clarifications are still needed. The experimental section and the way results are presented needs major improvement, as it is hard to tell if the attack is working and how efficient and effective it is from the data presented in this paper.



** Minor comments: **

* The Poison Frogs attack is described in Section 2, marking as a drawback the fact that it only works with fine-tuning. It is not clear however why this is a limitation, as one could train the model with normal training and add the poisoned data in the last epochs.

* In Algorithm 1, step 9: what is the update being performed? It seems to me that the pseudo-code does not capture the entire processing steps, hence making the whole work hard to reproduce.

* Figure 2 shows gradient alignment along epochs (please report the axis labels), however it does not seem "flat" in the end, it is slightly decreasing. What happens if we increase the number of epochs? Will the alignment disappear?

* Sometimes the reader's expertise is taken for granted (e.g. define "unrolled gradient"). This might make it difficult for the paper to reach a broader audience.

* Eq. 1 shows no constraints on the data points staying in the feature space after perturbations. Is it considered during the experiments?

* It is observed that VGG11 on CIFAR10 is less transferable, but it would be interesting to read a possible explanation for this phenomenon.

* Equations should distinguish vectors from scalars to improve readability.

* Figure 4 is unreadable as the text in labels and legends is too small.


** Comments after reading the authors' rebuttal **

I would like to thank the authors for their clarifications. The threat model is now clearer to me - and I think it deserves clarifications in the paper as well.

First of all, as far as I understand now, there's a net distinction between backdoors and clean-label attacks. Backdoor attacks assume that the attacker controls the design phase and the training process, and releases a backdoored model (which then someone else re-uses possibly with fine tuning). Hence, defenses against backdoors aim to detect whether models have been backdoored or not, and it is reasonable to expect that the defender doesn't know the training data as well as other design choices (as the attacker released the model). In this setting, clean-label attacks do not make sense (as the attacker controls the training labels too).

Clean-label attacks assume a different setup. Here the attacker only injects poisoning samples into the training set but does neither control the training process nor the training labels. Hence, clean-label attacks make sense in this setting. However, it also makes sense that the defender knows the training data (as the defender is the one that trains the algorithm, and the purpose is to either detect and remove the poisoning points or reduce their influence over training) - and hence I'm expecting the authors to do consider previous defenses that assume knowledge of the training set in their work.

To summarize, I think that:

(1) the authors should clarify in the title that they restrict themselves to clean-label integrity/targeted poisoning attacks.

(2) the authors should clarify the threat model, and clearly distinguish poisoning availability attacks (bilevel data poisoning) vs poisoning integrity attacks. Furthermore, in the poisoning integrity/targeted family, backdoor and clean-label attacks should be distinguished and the threat models clarified (in particular, w.r.t assumptions on what the attacker/defender know and have access to).

(3) the authors should revise their sentences on the complexity of data poisoning (previous clean-label targeted attacks like poison frogs are not as complex as bilevel data poisoning attacks). A fairer comparison in terms of complexity should also be considered - how faster is this new attack w.r.t. poison frogs and the other clean-label targeted attacks? (poisoning availability should not be considered here as the goal is different in that case).

(4) In general, there is need to disambiguate clean-label targeted poisoning attacks from the rest, and better position this work in context. Reading the paper in its current form, it seems that the authors are also able to improve scalability of poisoning availability attacks whereas this is not the goal of this work.

I'm willing to revise my score if the authors agree on making these clarifications in the paper, better highlighting the net contributions of their work and the proper context of competing approaches (which do not include backdoors and poisoning availability attacks).

---

> ### Author Response · Authors · 2020-11-13
> **Answer to Reviewer 4**
>
> **Different notions of data poisoning (Paragraph 3: “First, … caused by the attack”):** Thank you for your suggestions on nomenclature. We indeed refer to the definitions in Barreno et al, but follow the nomenclature used in recent  works of Shafahi et al and Huang et al.  We will make a point to clarify the notation and mention the poison integrity/ poison availability nomenclature..
>
> **Complexity in the setting of from-scratch victim training (Paragraph 4: “While this work … not mistaken”):** We will amend the sentence on the complexity of targeted attacks to specify the from-scratch setting. This is crucial because Koh et al. only consider a frozen feature extractor, simplifying the optimization problem significantly, compared to the “from-scratch” setting. Also, the DogFish dataset is simply 900 images of dogs and 900 of fish - this is not comparable to the >1,000,000 images in full ImageNet.
>
> **“From-scratch” setting (Paragraph 5: “Another important … fair enough”):**
> The attacker does not control the training routine of the victim. While the attacker can certainly use a pretrained network to craft poisons (as we do in our method), a victim will train a new network from a new random initialization - a setting where previous transfer based attacks fail - because they crucially rely on the feature extractor being fixed. These attacks break when the feature representation changes! We confirm the suspicion that these transfer based attacks do not succeed in the from-scratch setting in Table 2, where we replicate several previous methods, but train a victim from a new random initialization using the poisoned dataset.
>
>
> **“Clean-label” description of attacks (Paragraph 6: “There are parts … trigger on the image”):**
> * Clean-label attacks are more insidious, and realistic than label flipping attacks since clean-label attacks do not assume the attacker is also the labeler of the victim’s data. Many industrial practitioners will simply collect unlabeled data from the internet, and label it themselves (or employ services like Amazon Turk). Therefore an attacker cannot rely upon being able to incorrectly label any specific image. Moreover, targeted attacks become trivial in the label-flipping regime since the attacker could simply introduce the mislabeled target image into the victim’s training set.
> * Our attacks are still “clean-label” in the sense that these images are possibly noisy, but still undeniably images of e.g. dogs. Furthermore, the perturbations may be noticeable to a reader of the paper because we include the clean base images above, but an unwitting practitioner might not think twice about the poisoned images - the images pass a cursory glance from a human worker. Finally, we show our attack is successful for lower epsilon values, which are imperceptible perturbations. These can be found in fig. 3.
> * Designing a detector against adversarial attacks is actually surprisingly difficult, see (Carlini & Wagner, 2017 - cf. References section). That work discusses evasion attacks, however the same considerations hold in the targeted poisoning setting. We also test our attack against defenses that are meant to detect anomalies (see Fig 4).
> * We’d also like to point out that the considered threat model (small epsilon, small budget, clean-label attacks) has been an active field of past research on poisoning attacks against deep neural networks.
>
>
> **Defensive strategies (Paragraph 6:)**
> Thank you for pointing this out - yes, we only considered defenses that are known to work against targeted attacks and  we will clarify this in the revised version of our paper. For instance, the defense in Peri et al. successfully removes 100% of the poisons generated by poison frogs and convex polytopes. cf. Peri et al., and Hong et al. show that differential privacy reduces the effectiveness of Poison Frogs by 38.36%. Taking defenses that are specifically developed for targeted attacks on deep networks we compare to seems the most expressive numerical evaluation for our work. Furthermore, all but three of the defenses in Table 1 of https://arxiv.org/abs/1910.03137 require access to clean training data, an assumption that does not apply in our setting. These three remaining defenses (Tran et al., and Chen et al., Chen et al.) all rely upon the heuristic that poisons will be anomalous in feature space - an assumption we show does not apply for our attack. Furthermore, many of these defenses are in the setting of backdoor attacks, where a fixed, easily spotted patch is added to all poisons, not individually crafted perturbations as in our attack.
>
> **Hyperparameters and clean validation accuracy (Paragraph 7: “Finally, the experimental … of the model”)**
> Note that these parameters refer to variables introduced in Alg. 1. We will clarify this and backreference Alg. 1. Please see the above general comment regarding the definition of “poison success” and the natural validation accuracy of the poisoned models.

---

> > ### Comment · AnonReviewer4 · 2020-11-25
> > **Response to authors' rebuttal**
> >
> > Please refer to the updated review. Comments can be found at the end.

---

> ### Author Response · Authors · 2020-11-13
> **Answer to Reviewer 4 - Minor Comments**
>
> * “The Poison Frogs attack is described in Section 2, marking as a drawback the fact that it only works with fine-tuning. It is not clear however why this is a limitation, as one could train the model with normal training and add the poisoned data in the last epochs.”
> The attacker does not control how the victim trains their model. Poison Frogs works in the setting wherein the victim is using a transfer learning/fine-tuning training strategy, but not in the setting wherein a victim trains a randomly initialized model from scratch. Your proposed strategy is therefore not possible, firstly because it requires the attacker to gain knowledge of the new feature representation of the victim, and secondly because it requires the attacker to insert poisons in the middle of training. The attacker has no control over when poisons are inserted, they can only provide a modified dataset, as outlined in the threat model.
> * “In Algorithm 1, step 9: what is the update being performed? It seems to me that the pseudo-code does not capture the entire processing steps, hence making the whole work hard to reproduce.”
> The update being performed is one step of the first-order descent algorithm Adam, using the sign of the gradient, on the perturbation to the poisoned images where the objective is defined in eq. 3. We also include publicly available code in our submission to help reproduction of results.
> * “Figure 2 shows gradient alignment along epochs (please report the axis labels), however it does not seem "flat" in the end, it is slightly decreasing. What happens if we increase the number of epochs? Will the alignment disappear?”
> The alignment slowly decreases, but stays positive when increasing the number of epochs. This is a consequence of training. It is actually the bound (right-hand side of Eq.(4)).  that remains stable over additional epochs, see Fig. 5.
> * “Eq. 1 shows no constraints on the data points staying in the feature space after perturbations. Is it considered during the experiments?”
> While this could be a heuristic of the attacker, this is not a part of the targeted poisoning objective we consider. Note however that the feature representations of the poisons after training are not anomalous for their given class.

---

> ### Author Response · Authors · 2020-11-25
> **Answer to Final Comments - Both availability and integrity attacks are bilevel optimization problems**
>
> First off, we're very grateful for both your perspective and the substantial interest in clarifying this work - really.
> We will use this discussion to clarify our writing and improve our presentation of the taxonomy of poisoning attacks, and better differentiate the subfield of clean-label targeted poisoning attacks from other attack scenarios.
>
>
> **Some additional comments regarding points raised in the final comments:**
> * While we like the distinction between backdoor attacks and clean-label attacks proposed in this review, the general literature is sadly much less precise. Other works (and also we in the current version) use "backdoor" as synonym of "poison integrity". Under this definition our attack is also a backdoor attack. Our related work section differentiates backdoor - "poisoning attacks" like ours from "backdoor trigger attacks" (such as Saha2019)  based on the criterion that "trigger" attacks are allowed to modify both training and testing images [but not labels], whereas "poisoning attacks" may only modify images from the training set [but not labels].
>
> An orthogonal direction to this would be whether the attacker is allowed to modify not only the training set, but also provide a pretrained, but backdoored model, which implies partial control over the (pre)-training phase or control over access to training data. So clean-label attacks are backdoor attacks, depending on the definition.
> Our threat model in Sec. 3.1. defines the clean-label targeted-poisoning setting that we consider unambigously.
>
> * The defender in the "clean-label" setting (as defined in your second paragraph) is indeed allowed control over training data, but the crucial challenge for the defender is that there is no additional "clean training data" available. This makes defenses difficult as the defender has to assume that any data used for comparison has also been poisoned. As such, many defenses that rely on classification of poisoned and non-poisoned data fail because they have no basis of comparison. Only defenses based on unsupervised anomaly detection can still work in this setting - however many of these defenses are based on anomaly detection in feature space. Yet, we show that the feature space is non-anomalous after our attack (in Fig. 4a).
>
> * Important: Both "poison availability" and "poison integrity"/"targeted poisoning" are formally bilevel optimization problems. In both cases, the lower-level problem is the training of a model parametrized by theta with respect to some data x, whereas the higher-level problem is either to maximize the loss over held-out data in the case of poison availability, or to minimize the loss of some specific held-out target image in the case of targeted poisoning. For both, the higher level problem depends on model parameters theta which themselves depend on x, so that the overall objective can be optimized w.r.t to x.
> However in practice this objective is infeasible to solve and all attack schemes have consider heuristic or approximative solutions in some shape or form.
>
> * Poison Frogs is also a clean-label targeted attack, the only difference between our threat model and the threat model in Shafahi2018 is that there the feature extractor is kept fixed (and known to be fixed by both the attacker and the defender). MetaPoison is also a clean-label targeted attack, and it considers the same threat model that we consider (where the feature extractor is allowed to change and unknown to the attacker).
>
> * We will work on clarifying our threat model textually. As far as we understand currently though, the definition in 3.1 is unambiguous - if this is not the case, we would be very grateful for information where ambiguity remains.
>
> * We provide a complexity/effectiveness plot in Fig.10 which also includes poison frogs. Poison frogs has a similar complexity to our attack - however the attack fails in the from-scratch scenario, possibly because it does not approximat the bilevel objective well enough. In contrast, MetaPoison manages to approximate the objective better and leads to stronger attacks, but this comes at a signififcant computational complexity. Our attack is the first to attack the difficult scenario of from-scratch training with a method that is roughly as complex as poison frogs, but even stronger than MetaPoison. We will add additional clarification regarding this.
>
> * Note that we never compare to poison availability attacks in this work and do not run experiments in this setting. It is conceivable that the proposed gradient matching is also able to work in the poison availability setting (this would require the replacement of the currently considered target gradients with a negative gradient sample from the validation set), but we think this is a significantly different scenario that is better suited for future work.
>
>
> Again we're glad to have this discussion and will revise and clarify our work accordingly.

---

> > ### Comment · AnonReviewer4 · 2020-11-25
> > **Final remarks**
> >
> > Thanks for the prompt reply. Again, I think we substantially reached a good level of agreement.
> >
> > * I know that both integrity and availability poisoning attacks can be casted as a bilevel optimization. It only changes which points you select for the outer loss. However, this was not the main point of my review.
> >
> > I only think that the authors should work towards improving clarity. As they also recognize, the literature is quite confusing, and we have an opportunity for clarification. I suggest the authors to:
> >
> > * Clarify and narrow the scope of their attack immediately from the title and abstract. State that you are considering clean-label targeted/integrity attacks from the very beginning; and remove all the claims of large computational complexity which would better refer to availability attacks (e.g., in the abstract: "Previous poisoning attacks ... being prohibitively expensive for large datasets" - e.g., this is not the case for metapoisoning or poison frogs, which are your direct competitors).
> >
> > * Clarify your threat model in context. I think that the definition in 3.1 is fairly clear, even though some choices should be better motivated. In which practical cases the gray-box assumption that the architecture is known to the attacker makes sense? This should be motivated, and overall, it should be clarified how this attack makes different assumptions from other clean-label or backdoor poisoning attacks, if any, and why.
> >
> > * State that this attack may be also used in the context of poisoning availability attacks in future work (I fully agree with the last point of your answer).
> >
> > Overall, I really like to thank you for the fruitful discussions, and hope that my feedback can be useful to improve your work.

---

### Author Response · Authors · 2020-11-13
**General Comments**

We thank the reviewers for their constructive feedback. We will respond to specific points raised under their respective reviews. Here, we will respond to common concerns:

1) On the readability of figures, specifically Fig. 4: in all figures, we will make it more clear in the main body what “poison success”, or equivalently “poison accuracy” means. For convenience, this value refers to the percentage of runs (averaged over randomly initialized networks) in which the target image is mis-classified as the poison class, see the appendix for more details - we will make this clearer in the main text. As for the size of the text in Fig. 4, if one zooms in on a computer screen, the text becomes readable. However, we recognize this is an inconvenience, and not possible on a printed version, so we will make each subfigure its own figure, and expand the legend size.

2) The question of  validation accuracy of poisoned datasets on clean images was also a common concern. However validation accuracy is unaffected. Due to the considered threat model (small epsilon, small budget of 1%), the attack, as alluded to in the introduction, does not noticeably degrade the clean validation accuracy.
To emphasize this with actual data, we have included below the validation accuracy for the baseline experiments in the inset figure (subsection 5.1). These are the values for validation accuracy on CIFAR-10, for the poisoned dataset and the clean dataset:
* K=1,R=1: poisoned: 92.12, clean: 92.25
* K=2,R=1: poisoned: 92.06, clean: 92.16
* K=4,R=1: poisoned: 92.08, clean: 92.18
* K=8,R=1: poisoned: 92.20, clean: 92.16
* K=1,R=8: poisoned: 92.08, clean: 92.22
* K=2,R=8: poisoned: 92.03, clean: 92.27
* K=8,R=8: poisoned: 92.04, clean: 92.13

All values are averages over their respective runs. We will include a table with these natural accuracies in the updated appendix.

---

### Decision · Program_Chairs · 2021-01-07
**Final Decision**

**Decision:**

Accept (Poster)

**Comment:**

The paper presents a scalable data poisoning algorithm for targeted attacks, using the idea of designing poisoning patterns which "align" the gradients of the real objective and the adversarial objective. This intuition is supported by theoretical results, and the paper presents convincing experimental results about the effectiveness of the model.

The reviewers overall liked the paper. However, they requested a number of clarifications and some additional work, which should be incorporated in the final version (however, the authors are not required to use the wording as poison integrity/ poison availability). In particular, it would be great to see the experiment the authors suggested in their response to Reviewer 2 about the effectiveness of their method for multiple targets (this is important to better understand the limitations of the proposed approach).

---

> ### Comment · ~Jonas_Geiping1 · 2021-03-01
> **Final Version uploaded**
>
> Dear Program Chairs,
>
> we have uploaded a final version of this work, including additional experiments and discussion regarding multiple targets (See appendix F.8), a table showing clean validation accuracy (appendix F.9), and textual improvements to introduction and related work in direct response to many helpful suggestions from the reviewers - especially focusing on providing an improved understanding of the discussed attack within a wider taxonomy of data poisoning attacks.
>
> The implementation provided as supplementary material is maintained at https://github.com/JonasGeiping/poisoning-gradient-matching.